# CAUSALLY CORRECT PARTIAL MODELS FOR REINFORCEMENT LEARNING

## ABSTRACT

In reinforcement learning, we can learn a model of future observations and rewards, and use it to plan the agent's next actions. However, jointly modeling future observations can be computationally expensive or even intractable if the observations are high-dimensional (e.g. images). For this reason, previous works have considered partial models, which model only part of the observation. In this paper, we show that partial models can be causally incorrect: they are confounded by the observations they don't model, and can therefore lead to incorrect planning. To address this, we introduce a general family of partial models that are provably causally correct, yet remain fast because they do not need to fully model future observations.

## 1 INTRODUCTION

The ability to predict future outcomes of hypothetical decisions is a key aspect of intelligence. One approach to capture this ability is via *model-based reinforcement learning* (MBRL) (Munro, 1987; Werbos, 1987; Nguyen & Widrow, 1990; Schmidhuber, 1991). In this framework, an agent builds an internal representation $s_t$ by sensing an environment through observational data $y_t$ (such as rewards, visual inputs, proprioceptive information) and interacts with the environment by taking actions $a_t$ according to a policy $\pi(a_t|s_t)$. The sensory data collected is used to build a model that typically predicts future observations $y_{>t}$ from past actions $a_{\leq t}$ and past observations $y_{\leq t}$. The resulting model may be used in various ways, e.g. for planning (Oh et al., 2015; Silver et al., 2017a), generation of synthetic training data (Weber et al., 2017), better credit assignment (Heess et al., 2015), learning useful internal representations and belief states (Gregor et al., 2019; Guo et al., 2018), or exploration via quantification of uncertainty or information gain (Pathak et al., 2017).

Within MBRL, commonly explored methods include action-conditional, next-step models (Oh et al., 2015; Ha & Schmidhuber, 2018; Chiappa et al., 2017; Schmidhuber, 2010; Xie et al., 2016; Deisenroth & Rasmussen, 2011; Lin & Mitchell, 1992; Li et al., 2015; Diuk et al., 2008; Igl et al., 2018; Ebert et al., 2018; Kaiser et al., 2019; Janner et al., 2019). However, it is often not tractable to accurately model all the available information. This is both due to the fact that conditioning on high-dimensional data such as images would require modeling and generating images in order to plan over several timesteps (Finn & Levine, 2017), and to the fact that modeling images is challenging and may unnecessarily focus on visual details which are not relevant for acting. These challenges have motivated researchers to consider simpler models, henceforth referred to as *partial models*, i.e. models which are neither conditioned on, nor generate the full set of observed data (Guo et al., 2018; Gregor et al., 2019; Amos et al., 2018).

In this paper, we demonstrate that partial models will often fail to make correct predictions under a new policy, and link this failure to a problem in causal reasoning. Prior to this work, there has been a growing interest in combining causal inference with RL research in the directions of non-model based bandit algorithms (Bareinboim et al., 2015; Forney et al., 2017; Zhang & Bareinboim, 2017; Lee & Bareinboim, 2018; Bradtke & Barto, 1996) and causal discovery with RL (Zhu & Chen, 2019). Contrary to previous works, in this paper we focus on model-based approaches and propose a novel framework for learning better partial models. A key insight of our methodology is the fact that any piece of information about the state of the environment that is used by the policy to make a decision, but is not available to the model, acts as a confounding variable for that model. As a

result, the learned model is causally incorrect. Using such a model to reason may lead to the wrong conclusions about the optimal course of action as we demonstrate in this paper.

We address these issues of partial models by combining general principles of causal reasoning, probabilistic modeling and deep learning. Our contributions are as follows.

- We identify and clarify a fundamental problem of partial models from a causal-reasoning perspective and illustrate it using simple, intuitive Markov Decision Processes (MDPs) (Section 2).
- In order to tackle these shortcomings we examine the following question: What is the minimal information that we have to condition a partial model on such that it will be causally correct with respect to changes in the policy? (Section 4)
- We answer this question by proposing a family of viable solutions and empirically investigate their effects on models learned in illustrative environments (simple MDPs and 3D environments). Our method is described in Section 4 and the experiments are in Section 5.

## 2 A SIMPLE EXAMPLE: FUZZYBEAR

We illustrate the issues with partial models using a simple example. Consider the *FuzzyBear* MDP shown in Figure 1(a): an agent at initial state $s_0$ transitions into an encounter with either a teddy bear or a grizzly bear with $50\%$ random chance, and can then take an action to either hug the bear or run away. In order to plan, the agent may learn a partial model $q_\theta(r_2|s_0, a_0, a_1)$ that predicts the reward $r_2$ after performing actions $\{a_0, a_1\}$ starting from state $s_0$. This model is *partial* because it conditions on a sequence of actions without conditioning on the intermediate state $s_1$. The model is suitable for deterministic environments, but it will have problems on stochastic environments, as we shall see. Such a reward model is usually trained on the agent's experience which consists of sequences of past actions and associated rewards.

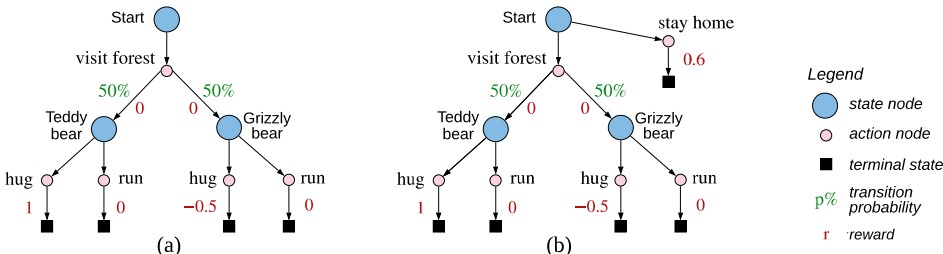

Figure 1: Examples of stochastic MDPs. (a) **FuzzyBear:** after visiting a forest, the agent meets either a teddy bear or a grizzly bear with $50\%$ chance and can either hug the bear or run away. (b) **AvoidFuzzyBear:** here, the agent has the extra option to stay home.

Now, suppose the agent wishes to evaluate the sequence of actions $\{a_0 = visit\ forest, a_1 = hug\}$ using the average reward under the model $q_\theta(r_2|s_0, a_0, a_1)$. From Figure 1(a), we see that the correct average reward is $0.5 \times 1 + 0.5 \times (-0.5) = 0.25$. However, if the model has been trained on past experience in which the agent has mostly hugged the teddy bear and ran away from the grizzly bear, it will learn that the sequence $\{visit\ forest, hug\}$ is associated with a reward close to $1$, and that the sequence $\{visit\ forest, run\}$ is associated with a reward close to $0$. Mathematically, the model will learn the following conditional probability:

$$p(r_2|s_0, a_0, a_1) = \sum_{s_1} p(s_1|s_0, a_0, a_1)p(r_2|s_1, a_1) = \sum_{s_1} \frac{p(s_1|s_0, a_0)\pi(a_1|s_1)}{\sum_{s_1'} p(s_1'|s_0, a_0)\pi(a_1|s_1')}p(r_2|s_1, a_1),$$

where $s_1$ is the state corresponding to either *teddy bear* or *grizzly bear*. In the above expression, $p(s_1|s_0, a_0)$ and $p(r_2|s_1, a_1)$ are the transition and reward dynamics of the MDP, and $\pi(a_1|s_1)$ is the agent's behavior policy that generated its past experience. As we can see, the behavior policy affects what the model learns.

The fact that the reward model $q_\theta(r_2|s_0, a_0, a_1)$ is not robust to changes in the behavior policy has serious implications for planning. For example, suppose that instead of visiting the forest, the agent

could have chosen to stay at home as shown in Figure 1(b). In this situation, the optimal action is to stay home as it gives a reward of $0.6$, whereas visiting the forest gives at most a reward of $0.5 \times 1 + 0.5 \times 0 = 0.5$. However, an agent that uses the above reward model to plan will overestimate the reward of going into the forest as being close to $1$ and choose the suboptimal action.[1]

One way to avoid this bias is to use a behavior policy that doesn't depend on the state $s_1$, i.e. $\pi(a_1|s_1) = \pi(a_1)$. Unfortunately, this approach does not scale well to complex environments as it requires an enormous amount of training data for the behavior policy to explore interesting states. A better approach is to make the model robust to changes in the behavior policy. Fundamentally, the problem is due to *causally incorrect reasoning*: the model learns the *observational conditional* $p(r_2|s_0, a_0, a_1)$ instead of the *interventional conditional* given by:

$$p(r_2|s_0, \mathrm{do}(a_0), \mathrm{do}(a_1)) = \sum_{s_1} p(s_1|s_0, a_0)p(r_2|s_1, a_1),$$

where the *do-operator* $\mathrm{do}(\cdot)$ means that the actions are performed *independently* of the unspecified context (i.e. independently of $s_1$). The interventional conditional is robust to changes in the policy and is a more appropriate quantity for planning. In contrast, the observational conditional quantifies the statistical association between the actions $a_0, a_1$ and the reward $r_2$ regardless of whether the actions caused the reward or the reward caused the actions. In Section 3, we review relevant concepts from causal reasoning, and based on them we propose solutions that address the problem.

Finally, although using $p(r_2|s_0, \mathrm{do}(a_0), \mathrm{do}(a_1))$ leads to causally correct planning, it is not optimal either: it predicts a reward of $0.25$ for the sequence $\{visit\,forest, hug\}$ and $0$ for the sequence $\{visit\,forest, run\}$, whereas the optimal policy obtains a reward of $0.5$. The optimal policy makes the decision after observing $s_1$ (teddy bear vs grizzly bear); it is *closed-loop* as opposed to *open-loop*. The solution is to make the intervention at the *policy* level instead of the *action* level, as we discuss in the following sections.

## 3 BACKGROUND ON CAUSAL REASONING

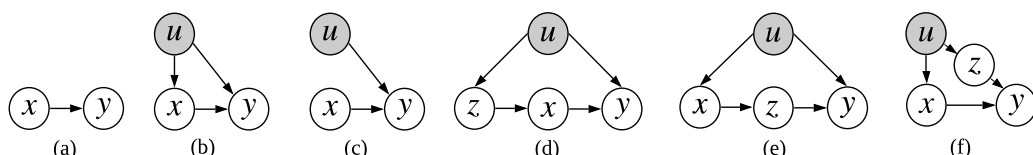

Figure 2: Illustration of various causal graphs. (a) Simple dependence without confounding. This is the prevailing assumption in many machine-learning applications. (b) Graph with confounding. (c) Intervention on graph (b) equivalent to setting the value of $x$ and observing $y$. (d) Graph with a backdoor $z$ blocking all paths from $u$ to $x$. (e) Graph with a frontdoor $z$ blocking all paths from $x$ to $y$. (f) Graph with a variable $z$ blocking the direct path from $u$ to $y$.

Many applications of machine learning involve predicting a variable $y$ (target) from a variable $x$ (covariate). A standard way to make such a prediction is by fitting a model $q_\theta(y|x)$ to a dataset of $(x, y)$-pairs. Then, if we are given a new $x$ and the data-generation process hasn't changed, we can expect that a well trained $q_\theta(y|x)$ will make an accurate prediction of $y$.

**Confounding:** In many situations however, we would like to use the data to make different kinds of predictions. For example, what prediction of $y$ should we make, if something in the environment has changed, or if we set $x$ ourselves? In the latter case $x$ didn't come from the original data-generation process. This may cause problems in our prediction, because there may be unobserved variables $u$, known as *confounders*, that affected both $x$ and $y$ during the data-generation process. That is, the actual process was of the form $p(u)p(x|u)p(y|x, u)$ where we only observed $x$ and $y$ as shown in Figure 2(b). Under this assumption, a model $q_\theta(y|x)$ fitted on $(x, y)$-pairs will converge to the target $p(y|x) \propto \int p(u)p(x|u)p(y|x, u)du$. However, if at prediction time we set $x$ ourselves, the actual distribution of $y$ will be $p(y|\mathrm{do}(x)) = \int p(u)p(y|x, u)du$. This is because setting $x$ ourselves changes the original graph from Figure 2(b) to the one in Figure 2(c).

---

[1]This problem is not restricted to toy examples. In a medical domain, a model could learn that leaving the hospital increases the probability of being healthy.

**Interventions:** The operation of setting $x$ to a fixed value $x_0$ independently of its parents, known as the *do-operator* (Pearl et al., 2016), changes the data-generation process to $p(u)\delta(x - x_0)p(y|x, u)$, where $\delta(x - x_0)$ is the delta-function. As explained above, this results in a different target distribution $\int p(u)p(y|x_0, u)du$, which we refer to as $p(y|\text{do}(x = x_0))$, or simply $p(y|\text{do}(x))$ when $x_0$ is implied. Let $\text{par}_j$ be the parents of $x_j$. The do-operator is a particular case of the more general concept of an *intervention*: given a generative process $p(x) = \prod_j p_j(x_j|\text{par}_j)$, an intervention is defined as a change that replaces one or more factors by new factors. For example, the intervention $p_k(x_k|\text{par}_k) \rightarrow \psi_k(x_k|\text{par}'_k)$ changes $p(x)$ to $p(x)\frac{\psi_k(x_k|\text{par}'_k)}{p_k(x_k|\text{par}_k)}$. The do-operator is a "hard" intervention whereby we replace a node by a delta function; that is, $p(x_{/k}, \text{do}(x_k = v)) = p(x)\frac{\delta(x_k - v)}{p_k(x_k|\text{par}_k)}$, where $x_{/k}$ denotes the collection of all variables except $x_k$.

**Backdoors and frontdoors:** In general, for graphs of the form of Figure 2(b), $p(y|x)$ does not equal $p(y|\text{do}(x))$. As a consequence, it is not generally possible to recover $p(y|\text{do}(x))$ using observational data, i.e. $(x, y)$-pairs sampled from $p(x, y)$, regardless of the amount of data available or the expressivity of the model. However, recovering $p(y|\text{do}(x))$ from observational data alone becomes possible if we assume additional structure in the data-generation process. Suppose there exists another observed variable $z$ that blocks all paths from the confounder $u$ to the covariate $x$ as shown in Figure 2(d). This variable is a particular case of the concept of a *backdoor* (Pearl et al., 2016, Chapter 3.3) and is said to be a backdoor for the pair $x - y$. In this case, we can express $p(y|\text{do}(x))$ entirely in terms of distributions that can be obtained from the observational data as:

$$p(y|\text{do}(x)) = \mathbb{E}_{p(z)}[p(y|z, x)]. \tag{1}$$

This formula holds as long as $p(x|z) > 0$ and is referred to as *backdoor adjustment*. The same formula applies when $z$ blocks the effect of the confounder $u$ on $y$ as in Figure 2(f). More generally, we can use $p(z)$ and $p(y|z, x)$ from Equation (1) to compute the marginal distribution $p(y)$ under an arbitrary intervention of the form $p(x|z) \rightarrow \psi(x|z)$ on the graph in Figure 2(b). We refer to the new marginal as $p_{\text{do}(\psi)}(y)$ and obtain it by:

$$p_{\text{do}(\psi)}(y) = \mathbb{E}_{\psi(x|z)p(z)}[p(y|z, x)]. \tag{2}$$

A similar formula can be derived when there is a variable $z$ blocking the effect of $x$ on $y$, which is known as a *frontdoor*, shown in Figure 2(e). Derivations for the backdoor and frontdoor adjustment formulas are provided in Appendix A.

**Causally correct models:** Given data generated by an underlying generative process $p(x)$, we say that a learned model $q_\theta(x)$ is causally correct with respect to a set of interventions $\mathcal{I}$ if the model remains accurate after any intervention in $\mathcal{I}$. That is, if $q_\theta(x) \approx p(x)$ and $q_\theta(x)$ is causally correct with respect to $\mathcal{I}$, then $q_{\theta, \text{do}(\psi)}(x) \approx p_{\text{do}(\psi)}(x)$ for all $\text{do}(\psi)$ in $\mathcal{I}$.

**Backdoor-adjustment and importance sampling:** Given a dataset of $N$ tuples $(z_n, x_n, y_n)$ generated from the joint distribution $p(u)p(z|u)p(x|z)p(y|x, u)$, we could alternatively approximate the marginal distribution $p_{\text{do}(\psi)}(y)$ after an intervention $p(x|z) \rightarrow \psi(x|z)$ by fitting a distribution $q_\theta(y)$ to maximize the re-weighted likelihood:

$$L(\theta) = \mathbb{E}_{p(u)p(z|u)p(x|z)p(y|x,u)}[w(x, z) \log q_\theta(y)] \approx \frac{1}{N}\sum_n w(x_n, z_n) \log q_\theta(y_n), \tag{3}$$

where $w(x, z) = \psi(x|z)/p(x|z)$ are the importance weights. While this solution is a mathematically sound way of obtaining $p_{\text{do}(\psi)}(y)$, it requires re-fitting of the model for any new $\psi(x|z)$. Moreover, if $\psi(x|z)$ is very different from $p(x|z)$ the importance weights $w(x, z)$ will have high variance. By fitting the conditional distribution $p(y|z, x)$ and using Equation (2) we can avoid these limitations.

**Connection to MBRL:** As we will see in much greater detail in the next section, there is a direct connection between partial models in MBRL and the causal concepts discussed above. In MBRL we are interested in making predictions about some aspect of the future (observed frames, rewards, etc.); these would be the dependent variables $y$. Such predictions are conditioned on actions which play the role of the covariates $x$. When using partial models, the models will not have access to the full state of the policy and so the policy's state will be a confounding variable $u$. Any variable in the computational graph of the policy that mediates the effect of the state in the actions will be a backdoor with respect to the action-prediction pair.

# 4 LEARNING CAUSALLY CORRECT MODELS

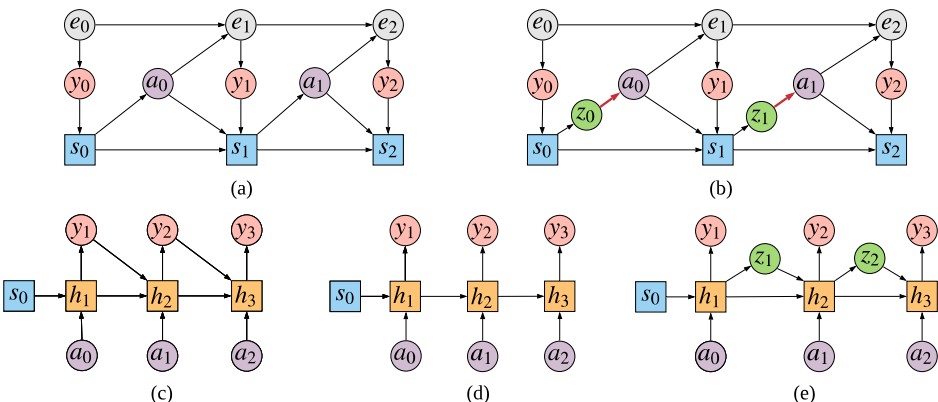

Figure 3: Graphical representations of the environment, the agent, and the various models. Circles are stochastic nodes, rectangles are deterministic nodes. (a) Agent interacting with the environment, generating a trajectory $\{y_t, a_t\}_{t=0}^T$. These trajectories are the training data for the models. (b) Same as (a) but also including the backdoor $z_t$ in the generated trajectory. The red arrows indicate the locations of the interventions. (c) Standard autoregressive generative model of observations. The model predicts the observation $y_t$ which it then feeds into $h_{t+1}$. (d) Example of a Non-Causal Partial Model (NCPM) that predicts the observation $y_t$ without feeding it into $h_{t+1}$. (e) Proposed Causal Partial Model (CPM), with a backdoor $z_t$ for the actions.

We consider environments with a hidden state $e_t$ and dynamics specified by an unknown transition probability of the form $p(e_t|e_{t-1}, a_{t-1})$. At each step $t$, the environment receives an action $a_{t-1}$, updates its state to $e_t$ and produces observable data $y_t \sim p(y_t|e_t)$ which includes a reward $r_t$ and potentially other forms of data such as images. An agent with internal state $s_t$ interacts with the environment via actions $a_t$ produced by a policy $\pi(a_t|s_t)$ and updates its state using the observations $y_{t+1}$ by $s_{t+1} = f_s(s_t, a_t, y_{t+1})$, where $f_s$ can for instance be implemented with an RNN. The agent will neither observe nor model the environment state $e_t$; it is a confounder on the data generation process. Figure 3(a) illustrates the interaction between the agent and the environment.

Consider an agent at an arbitrary point in time and whose current state[2] is $s_0$, and assume we are interested in generative models that can predict the outcome[3] $y_T$ of a sequence of actions $\{a_0, \ldots, a_{T-1}\}$ on the environment, for an arbitrary time $T$. A first approach, shown in Figure 3(c), would be to use an action-conditional autoregressive model of observations; initializing the model state $h_1$ to a function of $(s_0, a_0)$, sample $y_1$ from $p(.|h_1)$, update the state $h_2 = f_s(h_1, a_1, y_1)$, sample $y_2$ from $p(.|h_2)$, and so on until $y_T$ is sampled. In other words, the prediction of observation $y_T$ is conditioned on all available observations $(s_0, y_{<T})$ and actions $a_{<T}$. This approach is for instance found in (Oh et al., 2015).

In contrast, another approach is to predict observation $y_T$ given actions but using no observation data beyond $s_0$. This family of models, sometimes called *models with overshoot*, can for instance be found in (Silver et al., 2017b; Oh et al., 2017; Luo et al., 2019; Guo et al., 2018; Hafner et al., 2018; Gregor et al., 2019; Asadi et al., 2019) and is illustrated in Figure 3(d). The model deterministically updates its state $h_{t+1} = f_h(h_t, a_t)$, and generates $y_T$ from $p(.|h_T)$. An advantage of those models is that they can generate $y_T$ directly without generating intermediate observations.

More generally, we define a *partial view* $v_t$ as any function of past observations $y_{\leq t}$ and actions $a_{\leq t}$. We define a *partial model* as a generative model whose predictions are only conditioned on $s_0$, the partial views $v_{<t}$ and the actions $a_{<t}$: to generate $y_T$, the agent generates $v_1$ from $p(.|h_1)$, updates the state to $h_2 = f_h(h_1, v_1, a_1)$, and so on, until it has computed $h_T$ and sampled $y_T$ from $p(.|h_T)$. Both previous examples can be seen as special cases of a partial model, with $v_t = y_t$ and $v_t = \varnothing$ respectively.

---

[2]We reindex time for notational simplicity, but recall that $s_0$ is indeed a function of past observations $y_{\leq 0}$.

[3]For full generality, it may be that the predicted observation is only a subset or simple function of the full observation $y_T$; for instance one could predict only future rewards. For notational simplicity we make no difference between the full observation and the prediction.

Table 1: Comparison between noncausal partial model and the proposed architecture. The shaded cells indicate the key differences in architectures.

| | **NCPM architecture (overshoot)** | | | **CPM architecture** | | |
|---|---|---|---|---|---|---|
| **Agent** | | | | Backdoor | $z_t$ | $\sim m(z_t\|s_t)$ |
| | Action | $a_t$ | $\sim \pi(a_t\|s_t)$ | Action | $a_t$ | $\sim \pi(a_t\|z_t)$ |
| | State Update | $s_{t+1}$ | $= \mathrm{RNN}_s(s_t, a_t, y_{t+1})$ | State Update | $s_{t+1}$ | $= \mathrm{RNN}_s(s_t, a_t, y_{t+1})$ |
| **Model** | State Init. | $h_1$ | $= g(s_0, a_0)$ | State Init. | $h_1$ | $= g(s_0, a_0)$ |
| | | | | Backdoor | $z_t$ | $\sim p(z_t\|h_t)$ |
| | State Update | $h_{t+1}$ | $= \mathrm{RNN}_h(h_t, a_t)$ | State Update | $h_{t+1}$ | $= \mathrm{RNN}_h(h_t, z_t, a_t)$ |
| | Prediction | $y_t$ | $\sim p(y_t\|h_t)$ | Prediction | $y_t$ | $\sim p(y_t\|h_t)$ |

A subtle consequence of conditioning the model only on a partial view $v_t$ is that the variables $y_{<T}$ become confounders for predicting $y_T$, in addition to the state of the environment which is always a confounder. In Section 3 we showed that the presence of confounders makes it impossible to correctly predict the target distribution after changes in the covariate distribution. In the context of partial models, the covariates are the actions $a_{<T}$ executed by the agent and the agent's initial state $s_0$, whereas the targets are the predictions $y_T$ we want to make at time $T$. A corollary of this is that the learned partial model may not be robust against changes in the policy and thus cannot be used to make predictions under different policies $\pi$, and therefore should not be used for planning.

In Section 3 we saw that if there was a variable blocking the influence of the confounders on the covariates (a backdoor) or a variable blocking the influence of the covariates on the targets (a front-door), it may be possible to make predictions under a broad range of interventions if we learn the correct components from data, e.g. using the backdoor-adgustment formula in Equation (2). In general it may not be straightforward to apply the backdoor-adjustment formula because we may not have enough access to the graph details to know which variable is a backdoor. In reinforcement learning however, we can fully control the agent's graph. This means that *we can choose any node in the agent's computational graph that is between its internal state $s_t$ and the produced action $a_t$* as a backdoor variable for the actions. Given the backdoor $z_t$, the action $a_t$ is conditionally independent of the agent state $s_t$.

To make partial models causally correct, **we propose to choose the partial view $v_t$ to be equal to the backdoor $z_t$**. This allows us to learn all components we need to *make predictions under an arbitrary new policy*. In the rest of this paper we will refer to such models as *Causal Partial Models* (CPM), and all other partial models will be henceforth referred to as *Non-Causal Partial Models* (NCPM). We assume the backdoor $z_t$ is sampled from a distribution $m(z_t|s_t)$ and the policy is a distribution conditioned on $z_t$, $\pi(a_t|z_t)$. This is illustrated in Figure 3(b) and described in more details in Table 1(right). We can perform a simulation under a new policy $\psi(a_t|h_t, z_t)$ by directly applying the backdoor-adjustment formula, Equation (1), to the RL graph as follows:

$$p_{\mathrm{do}(\psi(a_t|h_t, z_t))}(y_{t+1}|h_t) = \mathbb{E}_{\psi(a_t|h_t, z_t)p(z_t|h_t)}[p(y_{t+1}|h_{t+1})], \qquad (4)$$

where the components $p(z_t|h_t)$ and $p(y_{t+1}|h_{t+1})$ with $h_{t+1} = f_h(h_t, z_t, a_t)$ can be learned from observational data produced by the agent.

Modern deep-learning agents (e.g. as in Espeholt et al. (2018); Gregor et al. (2019); Ha & Schmidhuber (2018)) have complex graphs, which means that there are many possible choices for the backdoor $z_t$. So an important question is: what are the simplest choices of $z_t$? Below we list a few of the simplest choices we can use and discuss their advantages and trade-offs; more choices for $z_t$ are listed in Appendix C.

**Agent state:** Identifying $z_t$ with the agent's state $s_t$ can be very informative about the future, but this comes at a cost. As part of the generative model, we have to learn the component $p(z_t|h_t)$. This may be difficult in practice when $z_t = s_t$ due to the high-dimensionality of $s_t$, hence and performing simulations would be computationally expensive.

**Policy probabilities:** The $z_t$ can be the vector of probabilities produced by a policy when we have discrete actions. The vector of probabilities is informative about the underlying state, if different states produce different probabilities.

**Intended action:** The $z_t$ can be the *intended* action before using some form of exploration, e.g. $\varepsilon$-greedy exploration. This is an interesting choice when the actions are discrete, as it is simple to model and, when doing planning, results in a low branching factor which is independent of the complexity of the environment (e.g. in 3D, visually rich environments).

The causal correction methods presented in this section can be applied to any partial model. In our experiments, we will focus on environment models of the form proposed by Gregor et al. (2019). These models consist of a deterministic "backbone" RNN that integrates actions and other contextual information. The states of this RNN are then used to condition a generative model of the observed data $y_t$, but the observations are not fed back to the model autoregressively, as shown in Table 1(left). This corresponds to learning a model of the form $p(y_t|s_0, a_0, \ldots, a_{t-1})$.

We will compare this against our proposed model, which allows us to simulate the outcome of any policy using Equation (4). In this setup, a policy network produces $z_t$ before an action $a_t$. For example, if the $z_t$ is the *intended* action before $\varepsilon$-exploration, $z_t$ will be sampled from a policy $m(z_t|s_t)$ and the *executed* action $a_t$ will then be sampled from an $\varepsilon$-exploration policy $\pi(a_t|z_t) = (1 - \varepsilon)\delta_{z_t, a_t} + \varepsilon\frac{1}{n_a}$, where $n_a$ is the number of actions and $\varepsilon$ is in $(0, 1)$. Acting with the sampled actions is diagrammed in Figure 3(b) and the mathematical description is provided in Table 1.

The model components $p(z_t|h_t)$ and $p(y_t|h_t)$ are trained via maximum likelihood on observational data collected by the agent. The partial model does not need to model all parts of the $y_t$ observation. For example, a model to be used for planning can model just the reward and the expected return. In any case, it is imperative that we use some form of exploration to ensure that $\pi(a_t|z_t) > 0$ for all $a_t$ and $z_t$ as this is a necessary to allow the model to learn the effects of the actions. The model usage is summarized in Algorithms 1 and 2 in Appendix D and we discuss the model properties in Appendix E.

# 5 EXPERIMENTS

We analyse the effect of the proposed corrections on a variety of models and environments. When the enviroment is an MDP, such as the FuzzyBear MDP from Section 2, we can compute exactly both the non-causal and the causal model directly from the MDP transition matrix and the behavior policy. In Section 5.1, we compare the optimal policies computed from the non-causal and the causal model via value iteration. For this analysis, we used the intended-action backdoor, since it's compatible with a tabular representation. In Section 5.2, we repeat the analysis using a learned model instead. For these experiments, we used the policy-probabilities backdoor. The optimal policies corresponding to a given model were computed using a variant of the Dyna algorithm (Sutton, 1991) or expectimax (Michie, 1966). Finally in Section 5.3, we provide an analysis of the model rollouts in a visually rich 3D environment.

## 5.1 VALUE-ITERATION ANALYSIS ON MDPS

Given an MDP and a behavior policy $\pi$, the optimal values $V_{M(\pi)}^*$ of planning based on a NCPM and CPM are derived in Appendix I. The theoretical analysis of the MDP does not use empirically trained models from the policy data, but rather assumes that the transition probabilities of the MDP and the policy from which training data are collected are accurately learned by the model. This allows us to isolate the quality of planning using the model from how accurate the model is.

**Optimal behavior policy:** The optimal policy of the FuzzyBear MDP (Figure 1(a)) is to always hug the teddy bear and run away from the grizzly bear. Using training data from this behavior policy, we show in Figure 7 (Appendix I) the difference in the optimal planning based on the NCPM (Figure 3(d)) and CPM with the backdoor $z_t$ being the *intended action* (Figure 3(e)). Learning from optimal policies with $\varepsilon$-exploration, the converged causal model is independent of the exploration parameter $\varepsilon$. We see effects of varying $\varepsilon$ on learned models in Figure 8 (Appendix I).

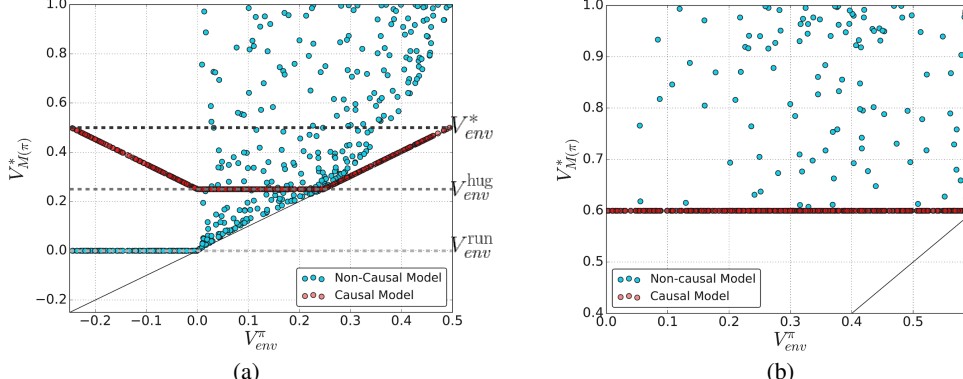

(a)                                             (b)

Figure 4: MDP Analysis: (a) In the FuzzyBear environment, we randomly generate 500 policies and scatter plot them with x-axis showing the quality of the behavior policy $V_{env}^\pi$ and y-axis showing corresponding model optimal evaluations $V_{M(\pi)}^*$. For each policy, we derive the corresponding converged model $M(\pi)$ equivalent to training on data generated by the policy. We then compute the optimal evaluation $V_{M(\pi)}^*$ using this model. We contrast the unrealistic optimism of the non-causal model evaluations $V_{\text{NCPM}(\pi)}^*$ with the more realistic causal model evaluations $V_{\text{CPM}(\pi)}^*$ for good policies $\pi$, as well as the over-pessimism of the non-causal model compared to the causal model for bad policies. (b) Same plot as (a) but for the AvoidFuzzyBear environment.

**Sub-optimal behavior policies:** We empirically show the difference between the causal and non-causal models when learning from randomly generated policies. For each policy, we derive the corresponding converged model $M(\pi)$ using training data generated by the policy. We then compute the optimal value of $V_{M(\pi)}^*$ using this model. On FuzzyBear (Figure 4(a)), we see that the causal model always produces a value greater than or equal to the value of the behavior policy. The value estimated by the causal model can always be achieved in the real environment. If the behavior policy was already good, the simulation policy used inside the model can reproduce the behavior policy by respecting the intended action. If the behavior policy is random, the intended action is uninformative about the underlying state, so the simulation policy has to choose the most rewarding action, independently of the state. And if the behavior policy is bad, the simulation policy can choose the opposite of the intended action. This allows to find a very good simulation policy, when the behavior policy is very bad. To further improve the policy, the search for better policies should be done also in state $s_1$. And the model can then be retrained on data from the improved policies.

If we look at the non-causal model, we see that it displays the unfortunate property of becoming more unrealistically optimistic as the behavior policy becomes better. Similarly, the worse the policy is, i.e. the lower $V_{env}^\pi$ is, the non-causal model becomes less able to improve the policy. On AvoidFuzzyBear (Figure 4(b)), the optimal policy is to stay at home. Learning from data generated by random policies, the causal model indeed always prefers to stay home with any sampled intentions, resulting in a constant evaluation for all policies. On the other hand, the non-causal model gives varied, overly-optimistic evaluations, while choosing the wrong action (visit forest).

## 5.2 LEARNED MODELS ON MDPS

We previously analyzed the case where the transition probabilities and theoretically optimal policy are known. We will now describe experiments with learned models trained by gradient descent, using the same training setup as described in Section 4.

**AvoidFuzzyBear with Dyna:** In this experiment we demonstrate that we can learn the optimal policy purely from off-policy experience using a general n-step-return algorithm derived from a causal model. The algorithm is described in detail in Appendix F. In short, we simulate experiences from the partial model, and use policy gradient to learn the optimal policy on these experiences as if they were real experiences (this is possible since the policy gradient only needs action probabilities, values, predicted rewards and ends of episodes). We compare a non-causal model and a causal

model where the backdoor $z_t$ is the intended action. For the environment we use AvoidFuzzyBear (Figure 1(b)). We collect experiences that are sub-optimal: half the time the agent visits the forest and half the time it stays home, but once in the forest it acts optimally with probability $0.9$. This is meant to simulate situations either where the agent has not yet learned the optimal policy but is acting reasonably, or where it is acting with a different objective (such as exploration or intrinsic reward), but would like to derive the optimal policy. We expect the non-causal model to choose the sub-optimal policy of visiting the forest, since the sequence of actions of visiting the forest and hugging typically yields high reward.

This is what we indeed find, as shown in Figure 5(a). We see that the non-causal model indeed achieves a sub-optimal reward (less than $0.6$), but believes that it will achieve a high reward (more than $0.6$). On the other hand, the causal model achieves the optimal reward and correctly predicts that it will achieve the corresponding value.

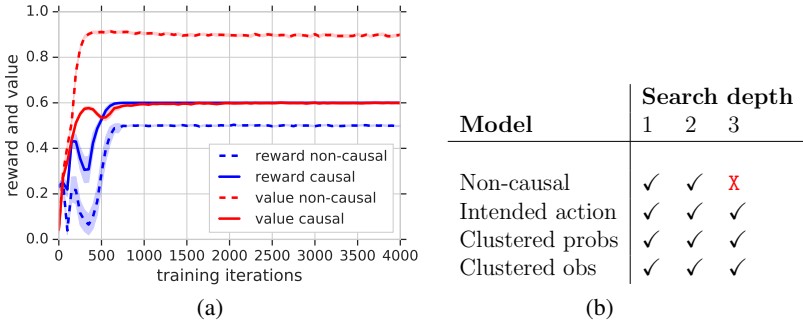

|  | (a) | | (b) | |
|---|---|---|---|---|

Figure 5: (a) Dyna on AvoidFuzzyBear. Models were trained on a sub-optimal behavior policy that explores both parts of the environment, and the evaluation policy was trained purely inside the model. The non-causal model (dotted lines) achieved sub-optimal reward, while expecting large reward. The causal model (solid lines) achieved optimal reward and had a correct expectation of the reward. The shaded area indicates 95% confidence intervals from 50 runs. (b) Models solving the AvoidFuzzyBear MDP with expectimax. The non-causal model misled the agent when using search depth 3 or higher.

**AvoidFuzzyBear with Expectimax:** In this experiment, we used the classical expectimax search (Michie, 1966; Russell & Norvig, 2009). On the simple AvoidFuzzyBear MDP, it is enough to use a search depth of 3: a decision node, a chance node and a decision node. The behavior policy was progressively improving as the model was trained.

In Figure 5(b), we see the results for the different models. Only the non-causal model was not able to solve the task. Planning with the non-causal model consistently preferred the stochastic path with the fuzzy bear, as predicted by our theoretical analysis with value iteration. The models with clustered probabilities and clustered observations approximate modeling of the probabilities or observations. These models are described in Appendix H.

## 5.3 VISUALLY RICH 3D ENVIRONMENT

The setup for these experiments is similar to Gregor et al. (2019), where the agent is trained using the IMPALA algorithm (Espeholt et al., 2018), and the model is trained alongside the agent via ELBO optimization on the data collected by the agent. The architecture of the agent and model is based on Gregor et al. (2019) and follows the description in Table 1(right). For these experiments, the backdoor $z_t$ was chosen to be the policy probabilities, and $p(z_t|h_t)$ was parametrized as a mixture of Dirichlet distributions. See Appendix J for more details. We demonstrate the effect of the causal correction on the 3D T-Maze environment where an agent walks around in a 3D world with the goal of collecting the reward blocks (food). The layout of this environment is shown in Figure 6(a). From our previous results, we expect NCPMs to be unrealistically optimistic. This is indeed what we see in Figure 6(b). Compared to NCPM, CPM with generated $z$ generates food at the end of a rollout with around 50% chance, as expected given that the environment randomly places the food on either side. In Figure 6(c) and Figure 6(d, left) we show subsets of rollouts from NCPM and CPM respectively (see Figures 10–12 in Appendix for full rollouts).

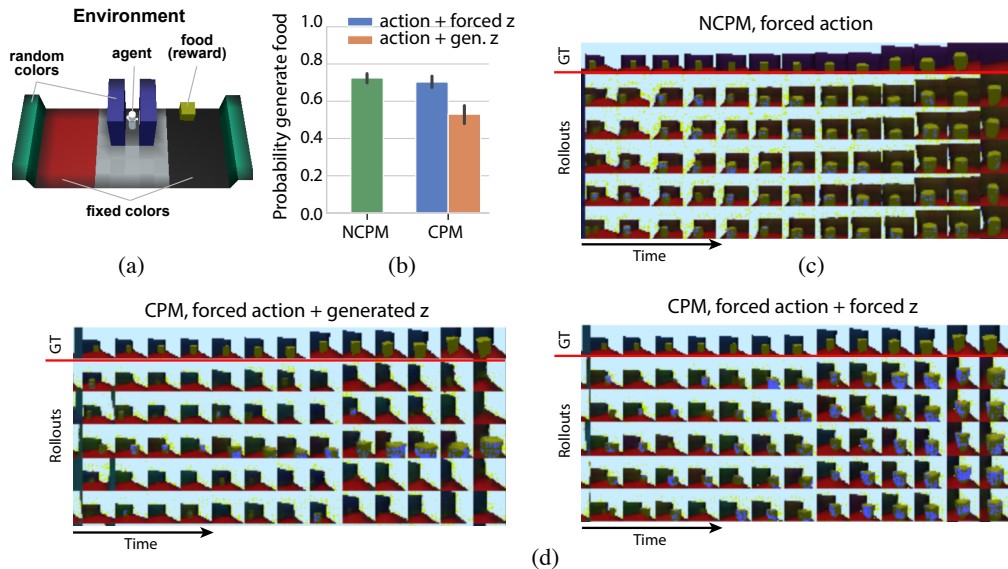

Figure 6: Causal partial model (CPM) and Non-causal partial model (NCPM) rollouts in a 3D T-Maze environment. (a) The agent always begins in a walled-off corridor, and can obtain food reward that spawns randomly on either the left (red) or the right (black) side. The colors of the corridor and side walls are randomized every episode. (b) Probability of the model generating food in rollouts for NCPM and CPM. Error bars represent standard error over 5 runs, each with 30 episodes. (c)-(d) Subset of frames from example rollouts using (c) NCPM and (d) CPM. In all rollouts depicted, the top row shows the real frames observed by an agent following a fixed policy (Ground Truth, GT). Bottom 5 rows indicate model rollouts, conditioned on 3 previous frames without revealing the location of the food. CPM and NCPM differ in their state-update formula and action generation (see Table 1), but frame generation $y_t \sim p(y_t|h_t)$ is the same for both, as introduced in Gregor et al. (2019). For CPM, we compare rollouts with forced actions and generated $z$ to rollouts with forced actions and forced $z$ from the ground truth. We can observe that rollouts with the generated $z$ (left) respect the randomness in the food placement (with and without food), while the rollouts with forced $z$ (right) consistently generate food blocks, if following actions consistent with the backdoor $z$ from the well-trained ground truth policy.

## 6 CONCLUSION

We have characterized and explained some of the issues of partial models in terms of causal reasoning. We proposed a simple, yet effective, modification to partial models so that they can still make correct predictions under changes in the behavior policy, which we validated theoretically and experimentally. The proposed modifications address the correctness of the model against policy changes, but don't address the correctness/robustness against other types of intervention in the environment. We will explore these aspects in future work.

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

## A  BACKDOOR AND FRONTDOOR ADJUSTMENT FORMULAS

Starting from a data-generation process of the form illustrated in Figure 2(b), $p(x, y, u) = p(u)p(x|u)p(y|x, u)$, we can use the do-operator to compute $p(y|\mathrm{do}(x)) = \int p(u)p(y|x, u)du$.

Without assuming any extra structure in $p(x|u)$ or in $p(y|x, u)$ it is not possible to compute $p(y|\mathrm{do}(x))$ from the knowledge of the joint $p(x, y)$ alone.

If there was a variable $z$ blocking all the effects of $u$ on $x$, as illustrated in Figure 2(d), then $p(y|\mathrm{do}(x))$ can be derived as follows:

$$\text{Joint density} \quad p(u)p(z|u)p(x|z)p(y|x, u) \tag{5}$$

$$\text{Intervention} \quad p(x|z) \to \psi(x) \tag{6}$$

$$\text{Joint after intervention} \quad p(u)p(z|u)\psi(x)p(y|x, u) \tag{7}$$

$$\text{Conditioning the new joint} \quad p(y|\mathrm{do}(x)) = \frac{\int p(u)p(z|u)\psi(x)p(y|x, u)dudz}{\psi(x)} \tag{8}$$

$$= \int p(z)p(y|x, z)dz \tag{9}$$

$$= \mathbb{E}_{p(z)}[p(y|x, z)], \tag{10}$$

where we used the formula

$$\int p(u)p(z|u)p(y|x, u)du = p(z)\int p(u|z)p(y|x, u)du \tag{11}$$

$$= p(z)p(y|x, z). \tag{12}$$

If instead of just fixing the value of $x$, we perform a more general intervention $p(x|z) \to \psi(x|z)$, then $p_{\mathrm{do}(\psi(x|z))}(y)$ can be derived as follows:

$$\text{Joint density} \quad p(u)p(z|u)p(x|z)p(y|x, u) \tag{13}$$

$$\text{Intervention} \quad p(x|z) \to \psi(x|z) \tag{14}$$

$$\text{Joint after intervention} \quad p(u)p(z|u)\psi(x|z)p(y|x, u) \tag{15}$$

$$\text{New marginal} \quad p_{\mathrm{do}(\psi(x|z))}(y) = \int p(u)p(z|u)\psi(x|z)p(y|x, u)dudzdx \tag{16}$$

$$= \int p(z)\psi(x|z)p(y|x, z)dzdx \tag{17}$$

$$= \mathbb{E}_{p(z)\psi(x|z)}[p(y|x, z)]. \tag{18}$$

Applying the same reasoning to the graph shown in Figure 2(e), we obtain the formula

$$p(y|\mathrm{do}(x)) = \mathbb{E}_{p(z|x)}[p(y|\mathrm{do}(z))] = \mathbb{E}_{p(z|x)p(x')}[p(y|x', z)], \tag{19}$$

where $p(z|x)$, $p(x')$ and $p(y|x', z)$ can be directly measured from the available $(x, y, z)$ data. This formula holds as long as $p(z|x) > 0, \forall x, z$ and it is a simple instance of *frontdoor adjustment* (Pearl et al., 2016).

## B  DERIVATION OF CAUSAL CORRECTNESS FOR THE MODELS IN FIGURE 3

Here, we will show in more detail that the models (c) and (e) in Figure 3 are causally correct, whereas model (d) is causally incorrect. Specifically, we will show that given an initial state $s_0$ and after setting the actions $a_0$ to $a_T$ to specific values, models (c) and (e) make the same prediction about the future observation $y_{T+1}$ as performing the intervention in the real world, whereas model (d) does not.

**Model (c)**   Using the do-operator, a hard intervention in the model is given by:

$$q_\theta(y_{T+1}|s_0, \text{do}(a_{0:T})) = q_\theta(y_{T+1}|s_0, a_{0:T}) = \int \prod_{t=1}^{T+1} q_\theta(y_t|h_t) \, dy_{1:T}, \tag{20}$$

where $h_t$ is a deterministic function of $s_0$, $a_{0:t-1}$ and $y_{1:t-1}$. The same hard intervention in the real world is given by:

$$p(y_{T+1}|s_0, \text{do}(a_{0:T})) = \int p(y_{1:T+1}|s_0, \text{do}(a_{0:T})) \, dy_{1:T} \tag{21}$$

$$= \int \prod_{t=1}^{T+1} p(y_t|s_0, \text{do}(a_{0:T}), y_{1:t-1}) \, dy_{1:T} \tag{22}$$

$$= \int \prod_{t=1}^{T+1} p(y_t|s_0, a_{0:t-1}, y_{1:t-1}) \, dy_{1:T}. \tag{23}$$

If the model is trained perfectly, the factors $q_\theta(y_t|h_t)$ will become equal to the conditionals $p(y_t|s_0, a_{0:t-1}, y_{1:t-1})$. Hence, an intervention in a perfectly trained model makes the same prediction as in the real world, which means that the model is causally correct.

**Model (d)**   The interventional conditional in the model is simply:

$$q_\theta(y_{T+1}|s_0, \text{do}(a_{0:T})) = q_\theta(y_{T+1}|s_0, a_{0:T}) = q_\theta(y_{T+1}|h_{T+1}), \tag{24}$$

where $h_{T+1}$ is a deterministic function of $s_0$ and $a_{0:T}$. In a perfectly trained model, we have that $q_\theta(y_{T+1}|h_{T+1}) = p(y_{T+1}|s_0, a_{0:T})$. However, the observational conditional $p(y_{T+1}|s_0, a_{0:T})$ is not generally equal to the inverventional conditional $p(y_{T+1}|s_0, \text{do}(a_{0:T}))$, which means that the model is causally incorrect.

**Model (e)**   Finally, the interventional conditional in this model is:

$$q_\theta(y_{T+1}|s_0, \text{do}(a_{0:T})) = q_\theta(y_{T+1}|s_0, a_{0:T}) = \int q_\theta(y_{T+1}|h_{T+1}) \prod_{t=1}^{T} q_\theta(z_t|h_t) \, dz_{1:T}, \tag{25}$$

where $h_t$ is a deterministic function of $s_0$, $a_{0:t-1}$ and $z_{1:t-1}$. The same intervention in the real world can be written as follows:

$$p(y_{T+1}|s_0, \text{do}(a_{0:T})) = \int p(y_{T+1}|s_0, \text{do}(a_{0:T}), z_{1:T}) p(z_{1:T}|s_0, \text{do}(a_{0:T})) \, dz_{1:T} \tag{26}$$

$$= \int p(y_{T+1}|s_0, \text{do}(a_{0:T}), z_{1:T}) \prod_{t=1}^{T} p(z_t|s_0, \text{do}(a_{0:T}), z_{1:t-1}) \, dz_{1:T} \tag{27}$$

$$= \int p(y_{T+1}|s_0, a_{0:T}, z_{1:T}) \prod_{t=1}^{T} p(z_t|s_0, a_{0:t-1}, z_{1:t-1}) \, dz_{1:T}. \tag{28}$$

In a perfectly trained model, we have that $q_\theta(y_{T+1}|h_{T+1}) = p(y_{T+1}|s_0, a_{0:T}, z_{1:T})$ and $q_\theta(z_t|h_t) = p(z_t|s_0, a_{0:t-1}, z_{1:t-1})$. That means that the intervention in a perfectly trained model makes the same prediction as the same intervention in the real world, hence the model is causally correct.

## C   ADDITIONAL CHOICES OF THE BACKDOOR $z_t$

The first alternative backdoor we consider is the empty backdoor:

**Empty backdoor** $z_t = \varnothing$**:** This backdoor is in general not appropriate; it is however appropriate when the behavior policy does in fact depend on no information, i.e. is not a function of the state

$s_t$. For example, the policy can be uniformly random (or any non-state dependent distribution over actions). This severely limits the behavior policy. Because the backdoor contains no information about the observations, the simulations are open-loop, i.e. we can only consider plans which consist of a sequence of fixed actions, not policies.

**An intermediate layer:** In principle, the $z_t$ can be any layer from the policy. To model the layer with a $p(z_t|h_t)$ distribution, we would need to know the needed numerical precision of the considered layer. For example, a quantized layer can be modeled by a discrete distribution. Alternatively, if the layer is produced by a variational encoder or variational information bottleneck, we can train $p(z_t|h_t)$ to minimize the $\mathrm{KL}(p_{encoder}(z_t|s_t) \parallel p(z_t|h_t))$.

Finally, if a backdoor is appropriate, we can combine it with additional information:

**Combinations:** It is possible to combine a layer with information from other layers. For example, the intended action can be combined with extra bits from the input layer. Such $z_t$ can be more informative. For example, the extra bits can hold a downsampled and quantized version of the input layer.

## D  ALGORITHMS FOR TRAINING AND SIMULATING FROM THE MODEL

Algorithms 1 and 2 describe how the model is trained and used to simulate trajectories. The algorithm for training assumes a distributed actor-learner setup (Espeholt et al., 2018).

---

**Algorithm 1** Model training

---

**Data collection on an actor:**
For each step:
    $z_t \sim m(z_t|s_t)$ ... sample the backdoor (e.g., the partial view with the intended action)
    $a_t \sim \pi(a_t|z_t)$ ... sample the executed action (e.g., add $\varepsilon$-exploration)
Collect:
    $s_t$ ... agent state
    $z_t$ ... partial view
    $a_t$ ... executed action
    $y_{t+1}$ ... targets (rewards, returns, ...)

**Model training on a learner:**
Require a trajectory: $s_0, a_{<T}, z_{<T}, y_{\leq T}$
$h_1 = g(s_0, a_0)$ ... initialize the model state
For each trajectory step:
    Train $p(y_t|h_t)$ to model $y_t$.
    Train $p(z_t|h_t)$ to model $z_t$.
    $h_{t+1} = \mathrm{RNN}_h(h_t, z_t, a_t)$ ... update the model state

---

---

**Algorithm 2** Using the model to generate a simulation under a new policy $\psi$

---

Require an agent state: $s_0$
$a_0 = \psi(a_0|s_0)$ ... choose the first action
$h_1 = g(s_0, a_0)$ ... initialize the model state
For each trajectory step:
    Predict the wanted targets $p(y_t|h_t)$ (e.g., rewards, returns, ...).
    $z_t \sim p(z_t|h_t)$ ... sample the partial view
    $a_t \sim \psi(a_t|h_t, z_t)$ ... choose the next action
    $h_{t+1} = \mathrm{RNN}_h(h_t, z_t, a_t)$ ... update the model state

---

## E  DISCUSSION OF MODEL PROPERTIES

Table 2 provides an overview of properties of autoregressive models, deterministic non-causal models and the causal partial models. The causal partial models have to generate only a partial view. The

partial view can be small and easy to model. For example, a partial view with the discrete intended action can be flexibly modeled by a categorical distribution.

The causal partial models are fast and causally correct in stochastic environments. The causal partial models have a low simulation variance, because they do not need to model and generate unimportant background distractors. If the environment has deterministic regions, the model can quickly learn to ignore the small partial view and collect information only from the executed action.

It is interesting that the causal partial models are invariant of the $\pi(a_t|z_t)$ distribution. For example, if the partial view $z_t$ is the intended action, the optimally learned model would be invariant of the used $\varepsilon$-exploration: $\pi(a_t|z_t)$. Analogously, the autoregressive models are invariant of the whole policy $\pi(a_t|s_t)$. This allows the autoregressive models to evaluate any other policy inside of the model. The causal partial model can run inside the simulation only policies conditioned on the starting state $s_0$, the actions $a_{<t}$ and the partial views $z_{\leq t}$. If we want to evaluate a policy conditioned on different features, we can collect trajectories from the policy and retrain the model. The model can always evaluate the policy used to produce the training data. We can also improve the policy, because the model allows to estimate the return for an initial $(s_0, a_0)$ pair, so the model can be used as a critic for a policy improvement.

Table 2: Models and their properties.

| | **Autoregressive** | **Deterministic** | **Causal Partial Model** |
|---|---|---|---|
| **Generates** | observation | nothing | partial view: $z_t$ |
| **Speed** | slow | **fast** | **fast** |
| **Causally correct** | **always** | in deterministic environments or with on-policy simulations | **always** |
| **Simulation variance** | high (distracted) | **lowest** | low |
| **Extra branching** | huge | **0** | controlled by $z_t$ size |
| **Invariant of** | $\pi(a_t|s_t)$ | - | $\pi(a_t|z_t)$ |
| **Evaluable policies** | **any** | $\psi(a_t|s_0, a_{<t})$ | $\psi(a_t|s_0, a_{<t}, z_{\leq t})$ |
| **Training** | **once** | iterative with policy | iterative with policy |

## F    DYNA-STYLE POLICY-GRADIENT ALGORITHM

In this section we derive an algorithm for learning an optimal policy given a (non-optimal) experience that utilizes n-step returns from partial models presented in this paper. In general, a model of the environment can be used in a number of ways for reinforcement learning. In Dyna (Sutton, 1990), we sample experiences from the model, and apply a model-free algorithm (Q-learning in the original implementation, but more generally we could consider SARSA or policy gradient) as if these were real experiences. In Dyna-2 (Silver et al., 2008), the same process is applied but in the context the agent is currently in—starting the simulations from the current state—and adapting the policy locally (for example through separate fast weights). In MCTS, the model is used to build a tree of possibilities. Can we apply our model directly in these scenarios? While we don't have a full model of the environment, we can produce a causally correct simulation of rewards and values; one that should generalize to policies different from those the agent was trained on. Policy probabilities, values, rewards and ends of episodes are the only variables that the above RL algorithms need.

Here we propose a specific implementation of Dyna-style policy-gradient algorithm based on the models discussed in the paper. This is meant as a proof of principle, and more exploration is left for future work.

As the agent sees an observation $y_{t+1}$, it forms an internal agent state $s_t$ from this observation and the previous agent state: $s_{t+1} = \text{RNN}_s(s_t, a_t, y_{t+1})$. The agent state in our implementation is the state of the recurrent network, typically LSTM (Hochreiter & Schmidhuber, 1997). Next, let us assume that at some point in time with state $s_0$ the agent would like to learn to do a simulation from the model. Let $h_t$ be the state of the simulation at time $t$. The agent first sets $h_1 = g(s_0, a_0)$ and proceeds with n-steps of the simulation recurrent network update $h_{t+1} = \text{RNN}(h_t, z_t, a_t)$. The agent learns the model $p(z_t|h_t)$ which it can use to simulate forward. We assume that the model was trained on some (non-optimal) policy/experience. We would like to derive an optimal policy and

value function. Since these need to be used during acting (if the agent were to then act optimally in the real environment), they are functions of the agent state $s_t$: $\pi(a_t|s_t), V(s_t)$. Now in general, $h_t \neq s_t$ but we would like to use the simulation to train an optimal policy and value function. Thus we define a second pair of functions $\pi_h(a_t|h_t, z_t), V_h(h_t, z_t)$. Here the extra $z_t$'s are needed, since the $h_t$ has seen $z$'s only up to point $z_{t-1}$.

Next we are going to train these functions using policy gradients on simulated experiences. We start with some state $s_t$ and produce a simulation $h_{t+1}, \ldots, h_T$ by sampling $z_t$ from the model at each step and action $a_t \sim \pi_h(a_t|h_t, z_t)$. However at the initial point $t$, we sample from $\pi$, not $\pi_h$, and compute the value $V$, not $V_h$. Sequence of actions, values and policy parameters are the quantities needed to compute a policy gradient update. We use this update to train all these quantities.

There is one last element that the algorithm needs. The values and policy parameters are trained at the start state and along the simulation by n-step returns, computed from simulated rewards and the bootstrap value at the end of the simulation. However this last value is not trained in any way because it depends on the simulated state $V_h(h_T)$ not the agent state $s_T$. We would like this value to equal to what the agent state would produce: $V(s_T)$. Thus, during training of the model, we also train $V_h(h_T)$ to be close to $V(s_T)$ by imposing an $L_2$ penalty. In our implementation, we actually impose a penalty at every point $t$ during simulation but we haven't experimented with which choice is better.

## G   SIMPLE EXTENSIONS

**Variance reduction.** To reduce the variance of a simulation, it is possible to sample the $z_t$ from a proposal distribution $q(z_t|h_t)$. The correct expectation can be still recovered by using an importance weight: $w = \frac{p(z_t|h_t)}{q(z_t|h_t)}$.

**Data efficient training.** Usually, we know the distribution of the used partial view: $z_t \sim m(z_t|s_t)$. When training the $p(z_t|h_t)$ model, we can then minimize the exact $\mathrm{KL}(m(Z_t|s_t) \parallel p(Z_t|h_t))$.

## H   MODELS TRAINED BY CLUSTERING

When using a tree-search, we want to have a small branching factor at the chance nodes. A good $z_t$ variable would be discrete with a small number of categories. This is satisfied, if the $z_t$ is the intended action and the number of the possible actions is small. We do not have such compact discrete $z_t$, if using as $z_t$ the observation, the policy probabilities or some other modeled layer. Here, we will present a model that approximates such causal partial models. The idea is to cluster the modeled layers and use just the cluster index as $z_t$. The cluster index is discrete and we can control the branching factor by choosing the the number of clusters.

Concretely, let's call the modeled layer $x_t$. We will model the layer with a mixture of components. The mixture gives us a discrete latent variable $z_t$ to represent the component index. To train the mixture, we use a clustering loss to train only the best component to model the $x_t$, given $h_t$ and $z_t$:

$$L_{\text{clustering}} = \min_{z_t}(-\beta_{\text{clustering}} \log p(z_t|h_t) - \log p(x_t|h_t, z_t)) \tag{29}$$

where $p(z_t|h_t)$ is a model of the categorical component index and $\beta_{\text{clustering}} \in (0, 1)$ is a hyper-parameter to encourage moving the information bits to the latent $z_t$. During training, we use the index of the best component as the inferred $z_t$. In theory, a better inference can be obtained by smoothing.

In contrast to training by maximum likelihood, the clustering loss uses just the needed number of the mixture components. This helps to reduce the branching factor in a search.

In general, the cluster index is not guaranteed to be sufficient as a backdoor, if the reconstruction loss $-\log p(x_t|h_t, z_t)$ is not zero. For example, if $x_t$ is the next observation, the number of mixture components may need to be unrealistically large, if the observation can contains many distractors.

# I  VALUE-ITERATION ANALYSIS ON MDPS

## I.1  OPTIMAL VALUE DERIVATIONS

We derive the following two model-based evaluation metrics for the MDP environments.

- $V^*_{\text{NCPM}(\pi)}(s_0)$: optimal value computed with the non-causal model, which is trained with training data from policy $\pi$, starting from state $s_0$.

- $V^*_{\text{CPM}(\pi)}(s_0)$: optimal value computed with the causal model, which is trained with training data from policy $\pi$, starting from state $s_0$.

The theoretical analysis of the MDP does not use empirically trained models from the policy data but rather assumes that the transition probabilities $p(s_{i+1} \mid s_i, a_i)$ of the MDP, and the policy, $\pi(a_i \mid s_i)$ or $\pi(z_i \mid s_i)$, from which training data are collected are accurately learned by the model.

**Computation of $V^*_{\text{NCPM}(\pi)}$ :**  For the non-causal model,

$$V^*_{\text{NCPM}(\pi)}(s_0) = \max_{a_0,\ldots,a_k} \sum_{i=0}^{k} \mathbb{E}_{s_i}\left[r_{i+1}(s_i, a_i) \mid s_0, a_0, a_1, \ldots, a_i\right]$$

$$= \max_{a_0,\ldots,a_k} \sum_{i=0}^{k} \sum_{s_i} p(s_i \mid s_0, a_0, \ldots, a_i) r_{i+1}(s_i, a_i).$$

Notice that the probability of $s_i$ is affected by $a_i$ here, because the network gets $a_i$ as an input, when predicting the $r_{i+1}$. This will introduce the non-causal bias. The network implements the expectation implicitly by learning the mean of the reward seen in the training data. We can compute the expectation exactly, if we know the MDP. The $p(s_i \mid s_0, a_0, \ldots, a_i)$ can be computed recursively in two-steps as:

$$p(s_i \mid s_0, a_0, \ldots, a_i) = \frac{p(s_i \mid s_0, a_0, \ldots, a_{i-1})\pi(a_i \mid s_i)}{\sum_{s'_i} p(s'_i \mid s_0, a_0, \ldots, a_{i-1})\pi(a_i \mid s'_i)}. \tag{30}$$

Here, we see the dependency of the learned model on the policy $\pi$. The remaining terms can be expressed as:

$$p(s_i \mid s_0, a_0, \ldots, a_{i-1}) = \sum_{s_{i-1}} p(s_i, s_{i-1} \mid s_0, a_0, \ldots, a_{i-1}) \tag{31}$$

$$= \sum_{s_{i-1}} p(s_{i-1} \mid s_0, a_0, \ldots, a_{i-1}) p(s_i \mid s_{i-1}, a_{i-1}). \tag{32}$$

Denoting $p(s_i \mid s_0, a_0, \ldots, a_j)$ by $S_{i,j}$, we have the two-step recursion

$$S_{i,i} = \frac{S_{i,i-1}\,\pi(a_i \mid s_i)}{\sum_{s'_i} S'_{i,i-1}\,\pi(a_i \mid s'_i)}, \tag{33}$$

$$S_{i,i-1} = \sum_{s_{i-1}} S_{i-1,i-1}\, p(s_i \mid s_{i-1}, a_{i-1}) \tag{34}$$

with $S_{1,0} = p(s_1 \mid s_0, a_0)$. We then compute $V^*_{\text{ncm}}(s_0)$ as $\max_{a_0,\ldots,a_k} \sum_{i=0}^{k} \sum_{s_i} S_{i,i} r_{i+1}(s_i, a_i)$.

**Computation of $V_{\text{CPM}(\pi)}^*$** :  For the causal model,

$$V_{\text{CPM}(\pi)}^*(s_0) = \max_{a_0} \sum_{z_1} p(z_1 \mid s_0, a_0) \max_{a_1} \sum_{z_2} p(z_2 \mid s_0, a_0, z_1, a_1) \cdots \tag{35}$$

$$\max_{a_{k-1}} \sum_{z_k} p(z_k \mid s_0, a_0, z_1, a_1, \ldots, z_{k-1}, a_{k-1}) \tag{36}$$

$$\max_{a_k} \sum_{i=0}^{k} \mathbb{E}[r_{i+1}(s_i, a_i) \mid s_0, a_0, z_1, a_1, \ldots, a_i)], \tag{37}$$

where for any $i \in [1, k]$,

$$p(z_i \mid s_0, a_0, z_1, a_1, \ldots, z_{i-1}, a_{i-1}) = \sum_{s_i} p(s_i, z_i \mid s_0, a_0, z_1, a_1, \ldots, z_{i-1}, a_{i-1}) \tag{38}$$

$$= \sum_{s_i} p(s_i \mid s_0, a_0, z_1 \ldots, z_{i-1}, a_{i-1}) \, \pi(z_i \mid s_i). \tag{39}$$

$$\tag{40}$$

Denoting $p(s_i \mid s_0, a_0, z_1 \ldots, z_{i-1}, a_{i-1})$ by $Z_i$, we have

$$Z_i = \sum_{s_{i-1}} p(s_{i-1}, s_i \mid s_0, a_0, z_1 \ldots, z_{i-1}, a_{i-1}) \tag{41}$$

$$= \sum_{s_{i-1}} p(s_i \mid s_{i-1}, a_{i-1}) p(s_{i-1} \mid s_0, a_0, z_1 \ldots, z_{i-1}, a_{i-1}) \tag{42}$$

$$= \sum_{s_{i-1}} p(s_i \mid s_{i-1}, a_{i-1}) p(s_{i-1} \mid s_0, a_0, z_1 \ldots, z_{i-1}), \tag{43}$$

where we used the fact that $s_{i-1}$ is independent of $a_{i-1}$, given $z_{i-1}$. Furthermore,

$$p(s_{i-1} \mid s_0, a_0, z_1 \ldots, z_{i-1}) = \frac{p(s_{i-1}, z_{i-1} \mid s_0, a_0, z_1 \ldots, a_{i-2})}{p(z_{i-1} \mid s_0, a_0, z_1 \ldots, a_{i-2})} \tag{44}$$

$$= \frac{\pi(z_{i-1} \mid s_{i-1}) p(s_{i-1} \mid s_0, a_0, z_1 \ldots, z_{i-2}, a_{i-2})}{\sum_{s'_{i-1}} \pi(z_{i-1} \mid s'_{i-1}) p(s'_{i-1} \mid s_0, a_0, z_1 \ldots, z_{i-2}, a_{i-2})} \tag{45}$$

$$= \frac{\pi(z_{i-1} \mid s_{i-1}) Z_{i-1}}{\sum_{s'_{i-1}} \pi(z_{i-1} \mid s'_{i-1}) Z'_{i-1}}. \tag{46}$$

Therefore we can compute $Z_i$ recursively,

$$Z_i = \sum_{s_{i-1}} p(s_i \mid s_{i-1}, a_{i-1}) \frac{\pi(z_{i-1} \mid s_{i-1}) Z_{i-1}}{\sum_{s'_{i-1}} \pi(z_{i-1} \mid s'_{i-1}) Z'_{i-1}} \tag{47}$$

with $Z_1 = p(s_1 \mid s_0, a_0)$. The last term to compute in the definition of $V^*_{\text{CPM}(\pi)}(s_0)$ is

$$\sum_{i=0}^{k} \mathbb{E}[r_{i+1}(s_i, a_i) \mid s_0, a_0, z_1, a_1, \dots, a_i)] = \sum_{i=0}^{k} \mathbb{E}[r_{i+1}(s_i, a_i) \mid s_0, a_0, z_1, a_1, \dots, z_i)] \quad (48)$$

$$= \sum_{i=0}^{k} \sum_{s_i} p(s_i \mid s_0, a_0, z_1, a_1, \dots, z_i) r_{i+1}(s_i, a_i) \quad (49)$$

$$= \sum_{i=0}^{k} \sum_{s_i} \frac{\pi(z_i \mid s_i) Z_i}{\sum_{s'_i} \pi(z_i \mid s'_i) Z'_i} r_{i+1}(s_i, a_i). \quad (50)$$

## I.2 PLANNING WITH NON-CAUSAL VS CAUSAL MODELS ON FUZZYBEAR

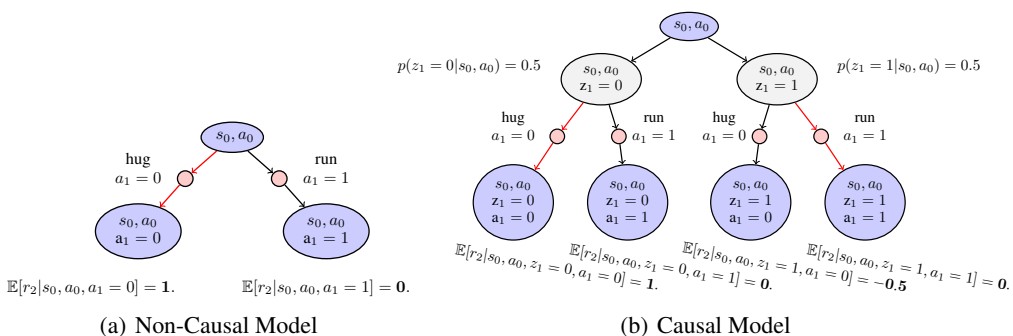

(a) Non-Causal Model

(b) Causal Model

Figure 7: FuzzyBear decision trees evaluated with NCPM and CPM based on the optimal policy data with the intended action. Decision paths through red action nodes that give the maximum expected rewards are highlighted in red. The blue nodes are states and the gray ones are chance nodes.

In Figure 7(a), the non-causal agent always chooses **hug** at step $t = 1$, since it has learned from the optimal policy that a reward of $+1$ always follows after taking $a_1 = $ hug. Thus from the non-causal agent's point of view, the expected reward is always 1 after hugging. This is wrong since only hugging a teddy bear gives reward 1. Moreover it exceeds the maximum expected reward $0.5$ of the FuzzyBear MDP. In Figure 7(b), the causal agent first samples the intention $z_1$ from the optimal policy, giving equal probability of landing in either of the two chance nodes. Then it chooses **hug** if $z_1 = 0$, indicating a teddy bear since the optimal policy intends to hug only if it observes a teddy bear. Likewise, it chooses **run** if $z_1 = 1$, indicating a grizzly bear. While the non-causal model expects unrealistically high reward, the causal model never over-estimates the expected reward.

## I.3 LEARNING WITH OPTIMAL POLICY AND VARYING $\varepsilon$-EXPLORATION:

We analyze learning from optimal policy with varying amounts of $\varepsilon$-exploration for models on FuzzyBear (Figure 8(a)) and AvoidFuzzyBear (Figure 8(b)). As the parameter $\varepsilon$-exploration varies in range $(0, 1]$, the causal model has a constant evaluation since the intended action is not affected by the randomness in exploration. The non-causal model, on the other hand, evaluates based on the deterministic optimal policy data (i.e. at $\varepsilon = 0$) at an unrealistically high value of $1.0$ when the maximum expected reward is $0.5$. As $\varepsilon \to 1$, the training data becomes more random and its optimal evaluation expectantly goes down to match the causal evaluation based on a uniformly random policy. The causal evaluation based on the optimal policy $V^*_{\text{CM}(\pi^*)}$ converges to the ground truth environmental evaluation $V^*_{env}$ as $\varepsilon \to 0$.

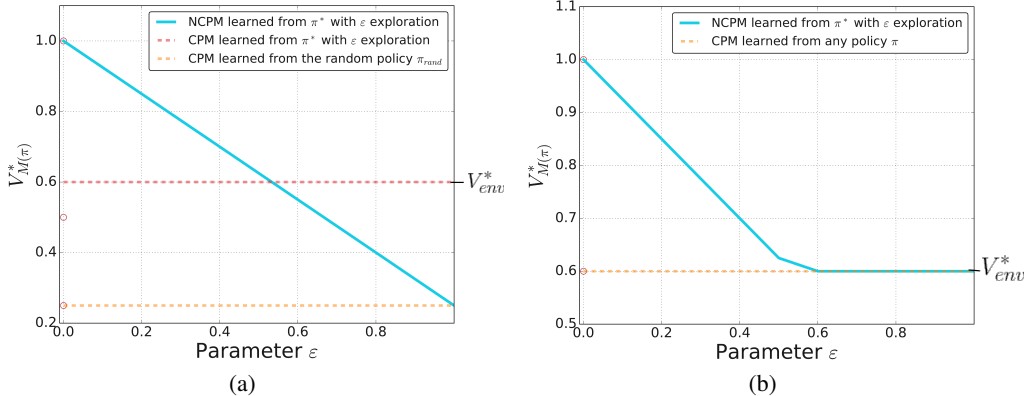

Figure 8: (a) In FuzzyBear, we use optimal policy with $\varepsilon$-exploration to generate training data for the Non-Causal Partial Model (NCPM) and Causal Partial Model (CPM). We vary the exploration parameter $\varepsilon \in (0, 1]$ and observe differences in found optimal values $V^*_{M(\pi)}$ under the model $M(\pi)$, where $M(\pi)$ denotes either CPM or NCPM trained on behavior policy $\pi$. The NCPM evaluation $V^*_{\text{NCPM}(\pi^*)}$ gives an unrealistically high value 1.0 learned from the deterministic optimal policy ($\varepsilon = 0$). Expectantly, it decreases to the level of CPM optimal value $V^*_{\text{CPM}(\pi_{\text{rand}})}$ learned from the uniformly random policy as $\varepsilon \to 1$. The CPM optimal values $V^*_{\text{CPM}(\pi^*)}$ are constant for any value of $\varepsilon$ based on the theoretical analysis in Section I.1. (b) shows the same plots as (a) for the AvoidFuzzyBear environment. Learning from any policy $\pi$, the CPM optimal value always equals the maximum expected reward 0.6, by correctly choosing to stay home.

## J  DETAILS FOR 3D EXPERIMENTS

### J.1  CONDITIONAL DIRICHLET MIXTURE

When the backdoor variable $z_t$ was chosen to be the action probabilities, the distribution $p(z_t|h_t)$ was chosen as a mixture-network with $N_c$ Dirichlet components. The concentration parameters $\alpha_k(h_t)$ of each component were parametrized as $\alpha_k(h_t) = \alpha \operatorname{softmax}(f_k(h_t))$, where $f_k$ is the output of a relu-MLP with layer sizes $[256, 64, N_c \times N_a]$, $\alpha$ is a total concentration parameter and $N_a$ is the number of actions.

### J.2  HYPER PARAMETERS AND TRAINING

The hyper-parameter value ranges used in our 3D experiments are similar to Gregor et al. (2019) and are shown in Table 3.

To speed up training, we interleaved training on the T-maze level with a simple "Food" level, in which the agent simply had to walk around and eat food blocks (described by Gregor et al. (2019)).

### J.3  ANALYSIS OF ROLLOUTS

For each episode, 5 rollouts are generated after having observed the first 3 frames from the environment. For the 5 rollouts, we processed the first 25 frames to classify the presence of food blocks by performing color matching of RGB values, using K-means and assuming 7 clusters. Rollouts were generated shortly after the policy had achieved ceiling performance (15–20 million frames seen), but before the entropy of the policy reduces to the point that there is no longer sufficient exploration. See Figure 9 for these same results for later training.

Table 3: Hyper-parameters used. Each reported experiment was repeated at least 5 times with different random seeds.

| Hyper-parameter | Description | Value |
|---|---|---|
| $\mu_{\text{policy}}$ | policy learning rate | 0.0001 |
| $\mu_{\text{model}}$ | model learning rate | 0.0005 |
| $c$ | policy entropy regularization | 0.0004 |
| $\beta_1$ | Adam $\beta_1$ | 0.9 |
| $\beta_2$ | Adam $\beta_2$ | 0.999 |
| $L_o$ | Overshoot Length | 8 |
| $L_u$ | Unroll Length | 100 |
| $N_t$ | Number of points used to evaluate the generative loss per trajectory | 6 |
| $N_g$ | Number of points used to evaluate the generative loss per overshoot | 2 |
| $N_s$ | Number of ConvDRAW Steps | 8 |
| $N_h$ | Number of units in LSTM | 512 |
| $\alpha$ | Total concentration of Dirichlet distributions | 100 |
| $N_c$ | Number of components of Dirichlet mixture | 10 |

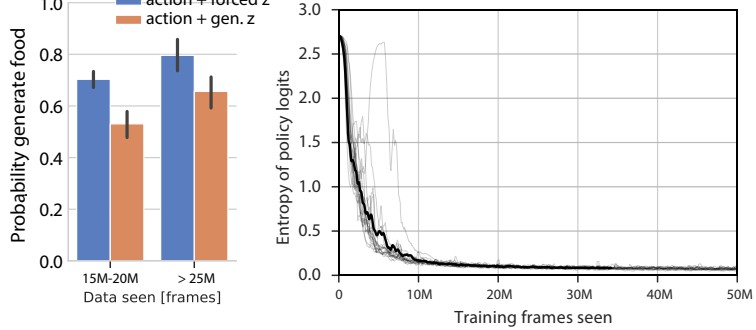

Figure 9: While earlier in training, CPM generates a diverse range of outcomes (food or no food), as the policy becomes more deterministic (as seen in the right plot of the policy entropy over training), CPM starts to generate more food and becomes overoptimistic, similar to NCPM. This can be avoided by training the model with non-zero $\varepsilon$-exploration.

# K   MODEL ROLLOUTS

Table 4: Different types of model rollouts considered. Blue cells indicate variables that require interaction with the real environment, depending on the agent's state $s_t$. Orange cells indicate variables that can be computed from the model's state $h_t$.

| | **Rollout** | **Rollout with forced actions** | **Rollout with forced actions and backdoor** |
|---|---|---|---|
| **Backdoor** | $z_t \sim p(z_t\|h_t)$ | $z_t \sim p(z_t\|h_t)$ | $z_t \sim m(z_t\|s_t)$ |
| **Action** | $a_t \sim \pi(a_t\|z_t)$ | $a_t \sim \pi(a_t\|\hat{z}_t); \hat{z}_t \sim m(\hat{z}_t\|s_t)$ | $a_t \sim \pi(a_t\|z_t)$ |
| **State** | $h_t = \mathrm{RNN}_h(h_{t-1}, a_{t-1}, z_{t-1})$ | | |
| **Prediction** | $y_t \sim p(y_t\|h_t)$ | | |

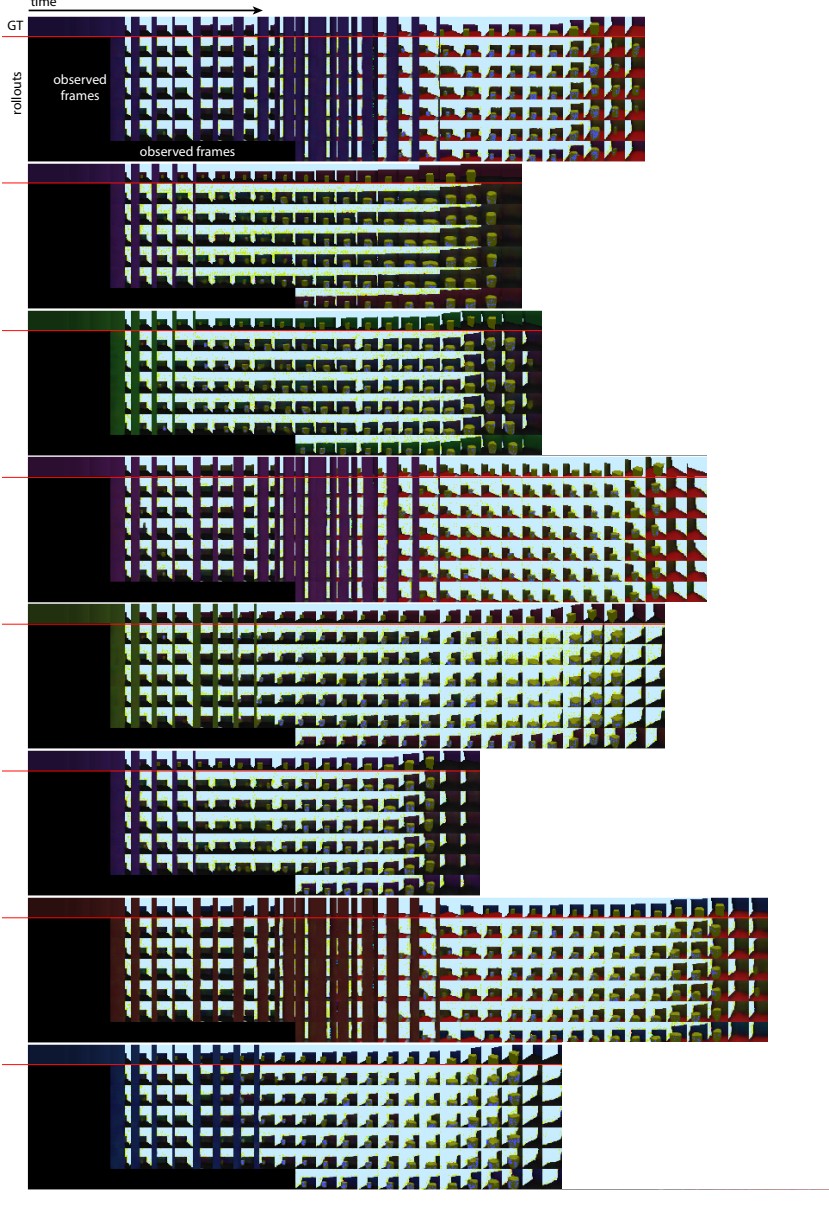

Figure 10: Full rollouts for NCPM, conditioned on forced actions.

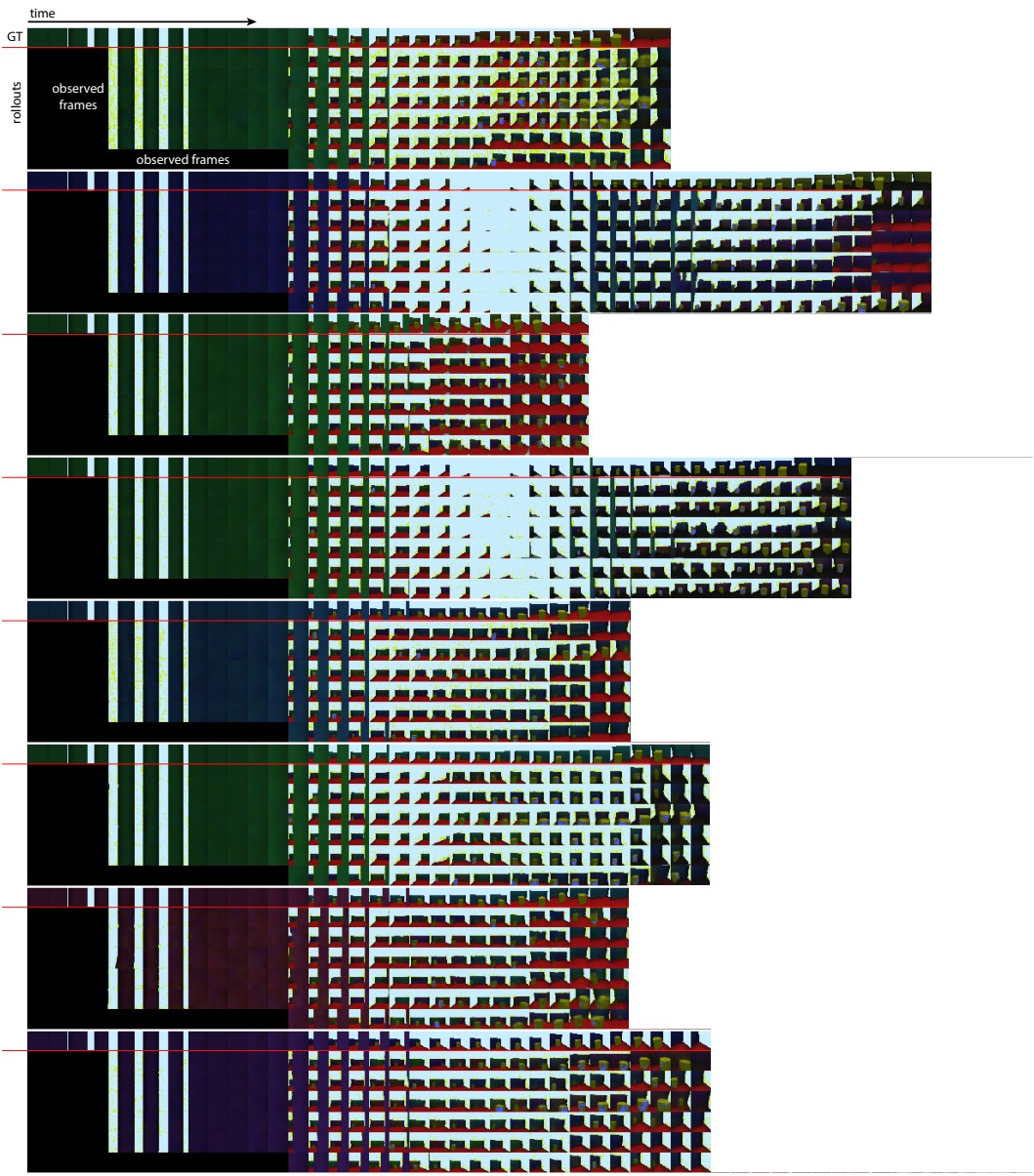

Figure 11: Full rollouts for CPM, conditioned on forced actions and generated $z$.

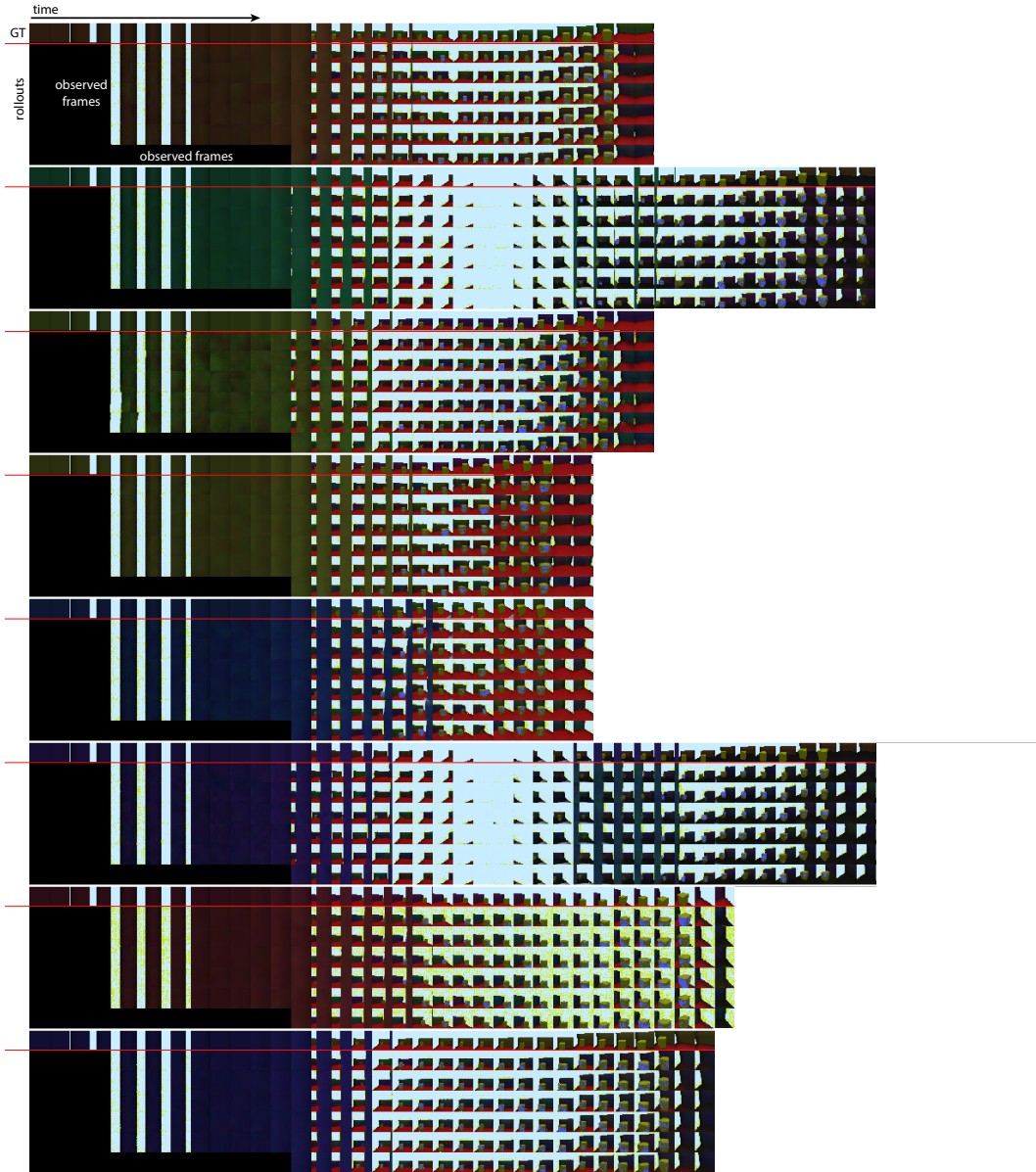

Figure 12: Full rollouts for CPM, conditioned on forced actions and forced $z$.

