# OpenReview forum: "Causally Correct Partial Models for Reinforcement Learning"
_ICLR.cc/2020/Conference — Reject_

### Official Review · AnonReviewer1 · 2019-10-23
**Official Blind Review #1**

**Rating:** 1

**Review:**

*Summary*

This paper considers the effect of partial models in RL, authors claim that these models can be causally wrong and hence result in a wrong policy (sub optimal set of actions). Authors demonstrate this issue with a simple MDP model, and emphasize the importance of behavior policy and data generation process. Then authors suggest a simple solution using backdoors (Pear et al) to learn causally correct models.They also conduct experiments to support their claims.

*Decision*

I vote for rejection of this paper, based on the following argument:

To my understanding authors are basically solving the “off-policy policy evaluation” problem, without relating to this literature. For example, the MDP example is just an off-policy policy evaluation problem, and it is very well known that in this case you need to consider the behavior policy, for example with importance sampling.
Even authors definition of the problem at the end of the page 4, and beginning of page 5, is the problem of “off-policy policy evaluation” when y_t = r_t
Authors have not cited any paper in this literature, and did not situate their work with respect to this literature. To my understanding, the proposed solution is basically importance sampling, that is very well known and studied in the field.

Additionally, I suggest that authors be more careful with their citations, for example, authors cited Silver et al 2017 [The Predictron: End-To-End Learning and Planning], as one recent paper using the method; however Silver et al 2017  is in MRP setting (Markov Reward process) where there is no action, so the described problem setting doesn't apply.

Improvement

The current manuscript needs a major revision, mainly 1) situate the work with respect to off-policy policy evaluation literature, and then 2) Considering step 1, a clarification for what is the novelty/ contribution of the current paper is needed.


**Experience Assessment:**

I have read many papers in this area.

**Review Assessment: Checking Correctness Of Derivations And Theory:**

I did not assess the derivations or theory.

**Review Assessment: Checking Correctness Of Experiments:**

I assessed the sensibility of the experiments.

**Review Assessment: Thoroughness In Paper Reading:**

I read the paper at least twice and used my best judgement in assessing the paper.

---

> ### Author Response · Authors · 2019-11-12
> **Authors response**
>
> We thank the reviewer for the constructive review.
>
> The MDP example from Section 2 is just for illustration purposes, it serves to demonstrate that learning an action-conditional partial model that has no access to the true state s will make the wrong predictions in general, resulting in sub-optimal behaviour.
>
> We understand that the proposed backdoor adjustment may seem similar to off-policy policy evaluation and we added an extra paragraph to Section 3 clarifying the difference between both ideas. For completeness, we discuss the key differences below.
>
> The main problem we are addressing is not only how to evaluate a new policy, but also how to learn a causally correct model that is robust against policy changes (this notion of robustness is more formally defined in Section 3). In order to make correct predictions under a new policy, the model must be trained in such a way as to learn the independent effects of the environment's stochasticity and the policy stochasticity (the actions) in the future states of the environment.
>
> In principle, this could be achieved with an importance-sampling method whereby we re-train the model with importance weights calculated using the ratio of old and new policy probabilities (it does not matter whether y=r or not). However, this solution suffers from two major drawbacks:
> 1) It requires either re-training the model for every new policy or keeping the entire dataset around, resulting in high computational cost.
> 2) As we grow the prediction sequence length, the importance weights quickly degenerate (effective sample size ~1), resulting in high-variance training.
>
> Our proposed solution via backdoor-adjustment allows us to make rollouts under a new policy *without re-training the model*. This is only possible because the model is broken down into two components:
> a) The prediction likelihood p(y | h, z, a) which is conditioned on the right variable, the backdoor z. This conditioning breaks the dependency between the environment's state and the policy's actions -- but only if z is a backdoor for the pair a-y.
> b) The likelihood p(z|h), which models the information necessary to act.
> These model components are independent of the behaviour policy \pi(a|z). We can thus replace \pi(a|z) by any other policy during evaluation/planning and the model rollouts should remain correct without any re-training or trajectory re-weighting.
>
> We believe these contributions constitute a substantial and qualitative deviation from the classical importance-sampling solution. Our objective is to learn a model suitable for planning (e.g. planning by tree search). Classical importance sampling cannot be used for this.
>
> We added a new discussion section to Appendix E comparing existing autoregressive models, deterministic models and the new causal partial models in the revised text. We hope that this will further clarify these points.
>
> Thank you for mentioning the citation Silver et al. (2017b). The revised text does not refer to it in the paragraph about action-conditional next-step models anymore.

---

### Official Review · AnonReviewer3 · 2019-10-23
**Official Blind Review #3**

**Rating:** 3

**Review:**

This paper tackles the issue of identifying the causal reasoning behind why partial models in MBRL settings fail to make correct predictions under a new policy. The novel contribution was a framework for learning better partial models based on models learning an interventional conditional, rather than an observational conditional. The paper tried to provide both theoretical and experimental reasoning for this framework.

I vote to (weak) reject the paper due to the major issues with section 2. Furthermore, the paper hard or at times almost impossible to understand as too many assumptions are made and too little is explained.

Recommendations

Because your graphs are not MDPs, you are not framing your example as an RL problem. This is causing a number of issues with notation and lack of clarity in the argument you're making.

1. It is unclear to me that the FuzzyBear example is correctly constructed as a RL example, reasons being that:
- Figure 1 (a) & (b) do not correspond to an MDP as two different states, teddy vs grizzly, are both designated as s_1 and similarly, the two possible actions, hug or run, are both designated as a_1 and thus are not distinct.
- Note your terminal state for the episodes
- Have a reward for (s0, a0) as every s-a pair should have a reward
- It would be helpful to note that the environments in Figure 1 are stochastic

2. Clarify notation. There are a number of assumptions about what background knowledg the reader should have. Given the bridging of disciplines in the paper, it would be useful to provide more detail on notation in Section 3.

3. Add a section on reinforcement learning in Section 3. If it's the last subsection in section 3, you could describe the relationship between the various causal reasoning and RL principles. This would further clarify how you're bridging these subtopics.

4. For sentence,

"Fundamentally, the problem is due to causally incorrect reasoning: the model learns the observational condi- tional p(r|s0, a0, a1) instead of the interventional conditional given by p(r|s0, do(a0), do(a1)) = s1 p(s1|s0, a0)p(r|s1, a1)."

As you don't cover the meaning of the do() operator until a later paragraph, provide a quick description of it as it is not common knowledge to a general AI audience, e.g., where do() indicates that the action was taken.

5. Correct the following sentences,

"Mathematically, the model with learn the following conditional probability:"

"In Section 3, we review relevant concepts from causal reasoning based on which we propose solutions that address the problem."

6. I recommend putting the interventional conditional equation, p(r|s0, do(a0), do(a1)) = 􏰀s1 p(s1|s0, a0)p(r|s1, a1), on its own line as the reader is doing a comparison of it with the previous equation, p(r|s0, a0, a1), given on page 2.

7. Strengthen your abstract by aligning more with claims you make in your conclusion.

8. The experiments in Figure 5 are averaged over 5 seeds. This is not enough to be statistically significant - furthermore, there are no error bars in the Figure.

Question(s):
1. You've indicated two policies for Figure 1 (a):
- pi1: the agent knows it is encountering a teddy bear, so it will hug
- pi2: the agent knows it is encountering a grizzly bear, so it will run
Is this the "change in the behavior policy" that you're referring to? If so, make this clearer, this currently requires a lot of work by the reader to make sense of it.

2. What are the partially observable parts of the environments in Figure 1 (a) & (b)? Make this clear.

**Experience Assessment:**

I have published one or two papers in this area.

**Review Assessment: Checking Correctness Of Derivations And Theory:**

I assessed the sensibility of the derivations and theory.

**Review Assessment: Checking Correctness Of Experiments:**

I assessed the sensibility of the experiments.

**Review Assessment: Thoroughness In Paper Reading:**

I read the paper at least twice and used my best judgement in assessing the paper.

---

> ### Author Response · Authors · 2019-11-12
> **Authors response**
>
> Thank you for your review. We substantially revised Section 2 based on your comments.
>
> 1. We’ve changed Figure 1 following your suggestions: we removed the confusing symbols s_1, a_1, etc. from the diagram, named each state and action uniquely, associated a reward with each state-action pair, clearly indicated the terminal state, and mentioned in the caption that the MDPs are stochastic. The new diagram should make the definition of the two FuzzyBear MDPs clear. As for the symbols s_1, a_1, etc., they should not be understood as designating specific states/actions, but as random variables ranging over states/actions. For example, s_1 is the random variable with the meaning “state at time 1” and can take on values in the set {teddy bear, grizzly bear}, where “teddy bear” and “grizzly bear” designate specific states.
>
> 2. We added a definition for the Dirac delta and the parent notation par. Are there any other notations you would like to be clarified?
>
> 3. We added a paragraph to Section 3 clarifying the connection to reinforcement learning.
>
> 4-7: We clarified and reworked writing based on your suggestions.
>
> 8. We reran the experiment with 50 seeds and provided 95% confidence intervals.
>
> Question 1: “Teddy bear” and “grizzly bear” are different states, so a policy is defined by the probability of the action “hug” (for instance) in each of those states. In other words, for this MDP, a policy is fully characterised by the pair (p1=P(hug | grizzly), p2=P(hug | teddy)). The change in behavior policy we consider is any change from a policy (p1, p2) used to collect data to an arbitrary policy (p1’, p2’).
>
> Question 2: This simple environment is not partially observable, only stochastic. We hope that the revised Figure 1 is much clearer.

---

### Official Review · AnonReviewer4 · 2019-11-02
**Official Blind Review #4**

**Rating:** 8

**Review:**

SUMMARY:
The authors apply ideas from causal learning to the problem of model learning in the context of sequential decision making problems. They show that models typically learned in this context can be problematic when used for planning. The authors then reformulate the model-learning problem using a causal learning framework and propose a solution to the above-mentioned problem using the concept of "backdoors." They perform experiments to demonstrate the advantages of doing so.

MAJOR COMMENTS:
Overall, I'm positive about the submission, though I think there's room for improvement.

The paper is well-written and does a very nice job of formulating the model-learning problem through the lens of causal learning, and it is convincing that the baseline model-learning procedure suffers from confounds. Moreover, the proposed modified procedure for model learning would seem to have the potential to fundamentally and positively impact model-learning in general. Finally, I found that limited experiments supported the points made in the paper.

That said, I think the paper as written could be substantially improved with a little extra effort. First, it does not -- in my opinion -- pass a basic reproducibility test, i.e., I am not confident that I could implement the proposed modified model-learning procedure even after reading the paper. The authors should seriously consider adding an algorithm box with pseudocode to both show the flow of training data to the model-learning steps and also how planning can be accomplished with the learned models.

Second, the paper would greatly benefit from any discussion (theoretical, experimental, or -- ideally -- both) of the tradeoffs that would arise when considering the proposed technique. What price is paid for the gain in planning accuracy? Is more data required for learning compared to non-causal approaches? If such a price is incurred, how was this made clear in the presented results?

Finally, an important aspect of the experiments seems to have gone un-discussed. Namely, what is the reason for the behavior of the proposed technique in Figure 4(a)? From much of the paper, I would have expected the red dots to live entirely on the horizontal dotted line $V^*_{env}$, but instead this only happens when the behavior policy has the same value. Why is this? Moreover, why do *better* behavior policies result in *worse* performance for the optimal model evaluation policies?

MINOR COMMENTS:
    * The authors should define "par_k", first referenced at the bottom of p3, before using it. Similarly for $\psi_k$
    * p5, second paragraph. I think there's a typo in "... sample $y_2$ from $q(\cdot | h_1)$ ..." (should probably be conditioned on $h_2$ not $h_1$).

POST-RESPONSE COMMENTS:
In my opinion, the authors adequately addressed both my own concerns, and also several valid concerns from the other reviewers. Therefore, I'm raising my score to "accept."

**Experience Assessment:**

I have read many papers in this area.

**Review Assessment: Checking Correctness Of Derivations And Theory:**

I assessed the sensibility of the derivations and theory.

**Review Assessment: Checking Correctness Of Experiments:**

I assessed the sensibility of the experiments.

**Review Assessment: Thoroughness In Paper Reading:**

I read the paper thoroughly.

---

> ### Author Response · Authors · 2019-11-12
> **Authors response**
>
> Thank you for the excellent review. Your comments greatly helped to improve the paper.
>
> 1) We  added two algorithms to the appendix, where we describe the model training and usage in a simulation.
>
> 2) We added an extra discussion section to the appendix, where we discuss the tradeoffs of different models. The manifested tradeoffs will depend on the used planning algorithms. For example, if using a tree search, the causal partial model introduces a larger branching factor and higher variance. In practice, we have not observed slower learning when using causal partial models. We will explore more advanced planning algorithms in future works.
>
> 3) Thank you for the curious questions. The Figure 4(a) is interesting exactly because of the observed behavior. In this experiment, the partial view is the intended action. If the behavior policy is random, the intended action is uninformative about the underlying state, so the simulation policy used inside the model has to choose the most rewarding action, independently of the state. If the behavior policy is good, the simulation policy can reproduce the behavior policy by respecting the intended action. And if the behavior policy is bad, the simulation policy can choose the opposite of the intended action. This allows us to find a very good simulation policy, when the behavior policy is very bad. Intuitively, given two options, a person who is consistently wrong is more informative than a person who is wrong 50% of the time.
>
> To further improve the policy, the search for better policies should be done also in state s_1. And the model can then be retrained on data from the improved policies. We updated the experiment description to clarify this.

---

> > ### Comment · AnonReviewer4 · 2019-11-15
> > **Response acknowledgement**
> >
> > Thanks to the authors for the response and for the edits to the paper. I'm very positive about the revisions that were made.
> >
> > Regarding Figure 4(a), I'm afraid I still find myself a bit confused. The author response seems to indicate that which behavior policy is used affects the quality of the simulation policy. However, it seemed to me that the whole point of the proposed technique was exactly to remove this effect.
> >
> > I suspect the subtlety lies in what is used as the partial view? If so, how should readers understand intuitively which partial views are better than others? Are there good ways to select the "right" partial view? Figure 4(a) seems to suggest that making a good choice is critical to the success of the proposed technique.

---

> > > ### Author Response · Authors · 2019-11-15
> > > **Response**
> > >
> > > Thank you for the response. That is a great question regarding the good choice of the partial view.
> > >
> > > First, to explain the invariance of the model: the behavior policy is the combination of the two arrows s_t -> z_t -> a_t as shown in Figure 3(b). The causally correct model is invariant with respect to a change in the arrow z_t -> a_t, shown as red in Figure 3(b), but not with respect to a change in s_t -> z_t. That's why the best-found simulation policy in Figure 4 changes with respect to the behavior policy. In any case however, the model will be causally correct, in the sense that it will evaluate correctly any simulation policy conditioned on the information available in the simulation: the s_0, the previous actions and the previous partial views.
> > >
> > > As you very correctly pointed out, the choice of the partial view z_t affects how good the simulation policy can be. In particular, what matters is how much information the partial view z_t has about the state s_t. If z_t has all the information about s_t, then the best-found simulation policy can be optimal, as shown in the leftmost and rightmost points of Figure 4(a). However, if z_t has no information about s_t, the best-found simulation policy will be the best open-loop policy, as shown in the middle of Figure 4(a).
> > >
> > > To choose a good partial view, the user can choose a sweet spot between the intended action and the full observation. A partial view with the intended action is enough to make the model causally correct. The intended action can be concatenated with more information from the observation, at the cost of increasing the variance of the simulation, increasing the branching factor of a tree search and making the partial view harder to model. In practice, the intended action worked surprisingly well.
> > >
> > > The best choice will depend on the environment and the planning algorithm. For example, if the observation background contains irrelevant noise, the partial view does not need to contain this noise. And if using just a small amount of planning before collecting more data, it is not needed to be able to express a simulation policy very different from the behavior policy. In the worst case, even if the planning is not able to find a better policy for actions a_1, a_2, ..., the model can be used to estimate the expected on-policy action-values for the (s_0, a_0) actions and improve the policy for a_0.
> > > We will explore multiple planning algorithms in future works.

---

### Official Review · AnonReviewer2 · 2019-11-04
**Official Blind Review #2**

**Rating:** 6

**Review:**

The paper considers the problem of predicting a variable y given x where x may suffer from a policy change, e.g., x may follow a different distribution than the original data or suffer from a confounding variable. The flow of the paper proceeds in learning a causally correct model in the sense that the model is robust to any intervention changes. Specifically, the paper considers a setting called "partial model" meaning a generative model conditioned on functions of past observations. To make the partial model causally correct, the paper considers the partial model conditioned on the backdoor that blocks all paths from the confounding variables.

1. The problem that this paper addresses seems to be new and interesting. The approach makes much sense: the problem is due to the confounding effect which can be addressed by introducing some other variables that implicitly blocks the confouders.

2. The  paper assumes the existence of the backdoor variable which is crucial for causal correctness. Does the backdoor always exist? Pearl's book may have some discussions on this. It will be still useful to include some materials in case of unfamiliar readers.



**Experience Assessment:**

I do not know much about this area.

**Review Assessment: Checking Correctness Of Derivations And Theory:**

N/A

**Review Assessment: Checking Correctness Of Experiments:**

I assessed the sensibility of the experiments.

**Review Assessment: Thoroughness In Paper Reading:**

I made a quick assessment of this paper.

---

> ### Author Response · Authors · 2019-11-12
> **Authors response**
>
> Thank you for the question.
>
> About the existence of the backdoor: It is true that in a general graphical model, backdoors may not always exist for a given pair of covariate/dependent variables. In reinforcement learning however, the backdoor always exists, because we have access to the entire computational graph of the policy. For example, the vector of the action probabilities can always serve as a backdoor. We revised the text to make this more clear.
>
> A case where we would not have access to a trivial backdoor would be, for example, when learning a policy from human demonstrations (since we do not have access to the human decision mechanisms). There, the partial view would need to be the observations experienced by the human.

---

### Author Response · Authors · 2019-11-15
**Summary of changes**

We thank all reviewers for their time and effort to improve the paper.
We substantially revised the paper and believe to have clarified all points of confusion.

The main changes to the paper:
1. Much cleaner Figure 1 with the MDP diagram.
2. Improved the overall clarity of Section 3.
3. Added a paragraph to Section 3 relating the backdoor adjustment to importance sampling.
4. Added another paragraph to Section 3 clarifying the relation between the causal concepts and RL.
5. Added Algorithm 1 to Appendix D to describe the model training.
6. Added Algorithm 2 with model usage to generate a simulation.
7. Added a discussion to Appendix E to compare the properties of autoregressive models, deterministic non-causal models and the causal partial models.
8. We reran the experiment from Figure 5 with 50 seeds (instead of 5) and provided 95% confidence intervals.

---

### Decision · Program_Chairs · 2019-12-19

**Decision:**

Reject

**Comment:**

The authors show that in a reinforcement learning setting, partial models can be causally incorrect, leading to improper evaluation of policies that are different from those used to collect the data for the model.  They then propose a backdoor correction to this problem that allows the model to generalize properly by separating the effects of the stochasticity of the environment and the policy.  The reviewers had substantial concerns about both issues of clarity and the clear, but largely undiscussed, connection to off-policy policy evaluation (OPPE).

In response, the authors made a significant number of changes for the sake of clarity, as well as further explained the differences between their approach and the OPPE setting.  First, OPPE is not typically model-based.  Second, while an importance sampling solution would be technically possible, by re-training the model based on importance-weighted experiences, this would need to be done for every evaluation policy considered, whereas the authors' solution uses a fundamentally different approach of causal reasoning so that a causally correct model can be learned once and work for all policies.

After much discussion, the reviewers could not come to a consensus about the validity of these arguments.  Futhermore, there were lingering questions about writing clarity. Thus, in the future, it appears the paper could be significantly improved if the authors cite more of the off policy evaluation literature, in addition to their added textual clairifications of the relation of their work to that body of work.  Overall, my recommendation at this time is to reject this paper.